# Limits to Predicting Online Speech Using Large Language Models

## Abstract

Our paper studies the predictability of online speech – that is, how well language models learn to model the distribution of user-generated content on X (previously Twitter). We define predictability as a measure of the model's uncertainty, i.e., its negative log-likelihood. As the basis of our study, we collect 10M tweets for "tweet-tuning" base models and a further 6.25M posts from more than five thousand X (previously Twitter) users and their peers. In our study involving more than 5000 subjects, we find that predicting posts of individual users remains surprisingly hard. Moreover, it matters greatly what context is used: models using the users' own history significantly outperform models using posts from their peers. We validate these results across four large language models ranging in size from 1.5 billion to 70 billion parameters. Moreover, our results replicate if instead of prompting the model with additional context, we finetune on it. We follow up with a detailed investigation on what is learned in-context and a demographic analysis. Between 4% and 20% of what is learned in-context is the use of @-mentions and hashtags, depending on the type of context. Our main results hold across the demographic groups we studied.

## 1 Introduction

Prediction is of fundamental importance for social research Salganik (2019); Salganik et al. (2020). The predictability of different social variables can provide a scientific window on a diverse set of topics, such as emotion contagion Kramer et al. (2014) and social influence Bagrow et al. (2019); Cristali & Veitch (2022); Qiu et al. (2018), privacy concerns Garcia (2017); Garcia et al. (2018); Li et al. (2012); Jurgens et al. (2017), the behavior of individuals and groups Tyshchuk & Wallace (2018); Nwala et al. (2023), the heterogeneity of networks Colleoni et al. (2014); Aiello et al. (2012), information diffusion Chen et al. (2019); Weng et al. (2014); Bourigault et al. (2014); Guille & Hacid (2012) and more. Of particular importance for the study of digital platforms is the case of online speech. Understanding and modeling language usage on social media Kern et al. (2016); Schwartz et al. (2013) has become of particular interest to the research community, where Twitter is one of the most studied social media platforms Zhang et al. (2023); Qudar & Mago (2020).

We revisit the problem of predicting online speech in light of dramatic advances in language modeling. We focus on the central question: How predictable is our online speech using large language models? Such predictive capabilities could inform research on substantial risks – such as user profiling, impersonation and exerting influence on real users Carroll et al. (2023); Weidinger et al. (2022). Inspired by work exploring the possibility of user profiling through their peers Bagrow et al. (2019), we ask the following: How predictable is a social media post given posts from the author's peers? We contrast the answer with how predictable social media posts are from the authors' own posts. Through experiments spanning millions of tweets and thousands of subjects, our study provides a detailed picture of the current state of predicting online speech and the potential risks that stem from it.

### 1.1 Contributions

We investigate the predictability of online speech on the social media platform X (Twitter) using a corpus of 6.25M posts (tweets) of 5000 subjects and their peers. An additional 10M tweets were reserved for

tweet-tuning models. We test four large language models of increasing size: GPT-2-XL-1.5B, Llama-3-8B, Falcon-40B, and Llama-2-70B. We use these models to estimate the predictability of our subjects' posts under various settings. We vary the type of context we provide to our model and observe the effect on model uncertainty. We study the following settings: *no context*, *random context*, *peer context* and *user context*. We present a detailed analysis on what's learned in-context and the robustness of our findings. Furthermore, we explore how our results apply to different demographic groups. Our contributions can be summarized as follows:

- **LLMs struggle to predict online speech** In Section 4.4, we show that most users' posts are less predictable than posts from financial news accounts. However, even with additional context prediction performs relatively poorly. Only our largest model with additional user context (see Llama-2-70B with user context in Fig. 1) is able to approximate the estimated entropy rate of the English language (1.12 bits). Subjects from Nigeria are the least predictable (Section 4.5). Between 4% and 20% of the effect size can be attributed to the model learning to predict hashtags and @-mentions (i.e. syntax), depending on the type of context (Section 4.2).

- **Predictability depends largely on context** In Section 4.1 we show that a user's own posts have significantly more predictive information than posts from their most frequently @-mentioned accounts. Prediction benefits most from user context, followed by peer context, in turn followed by random context. Our prompting experiments in Figure 1 illustrate these findings, which are robust to model choice and evaluation strategy and even replicate across different demographic groups (Section 4.5). We find that all types of context improve predictability significantly, with a large effect size (Fig. 3). Our finetuning experiments from Section 4.3 suggest that whatever predictive information is inside the peer context can also be found in the user context.

Our results on the predictability of online speech may inspire research on a wide range of topics. These may include research on influence and information propagation on social networks, social homophily and potential risks. To summarize, the extent to which we can predict online speech is limited even with the large language models we study. Our observations do not suggest that peers exert an outsize influence on an individual's online posts. Concerns that large language models have made online speech broadly predictable are not supported by our findings, though we do observe substantial variability at the individual level.

## 2 Related work

**Modeling online speech using language models** Understanding and modeling language usage on social media Bashlovkina et al. (2023); Kern et al. (2016); Schwartz et al. (2013) has become of particular interest to the research community, where Twitter is one of the most studied social media platforms Zhang et al. (2023); Qudar & Mago (2020). Many works focus on LLMs' ability to predict singular, sensitive attributes of users based on what they post. Users' gender, location and relationship status Staab et al. (2023) - even users' political leaning Jiang et al. (2023), morality or toxicity Jiang & Ferrara (2023) are predictable. Our central question is: How predictable is our *online speech* using large language models? Such predictive capabilities could inform research on substantial risks – such as user profiling, impersonation and exerting influence on real users Carroll et al. (2023); Weidinger et al. (2022). We do this through the lens of model uncertainty, which gives us a natural information-theoretic interpretation of our results. We use the cross-entropy of the English language Shannon (1951); Takahashi & Tanaka-Ishii (2018) as a baseline[1].

**Prediction based on neighbors** Predicting attributes of a node from its neighbors is a well-known paradigm in machine learning. In the context of social networks, however, it often has problematic implications on privacy since it limits the user's ability to control what can be inferred about them Garcia (2017); Garcia et al. (2018); Li et al. (2012). Work on the feasibility of "shadow profiles" (predicting attributes of non-users from platform users) has introduced a notion of privacy that is collective Garcia (2017); Garcia

---

[1]Of course while keeping in mind that this is an imperfect comparison. Social media language differs from conventional language Bashlovkina et al. (2023).

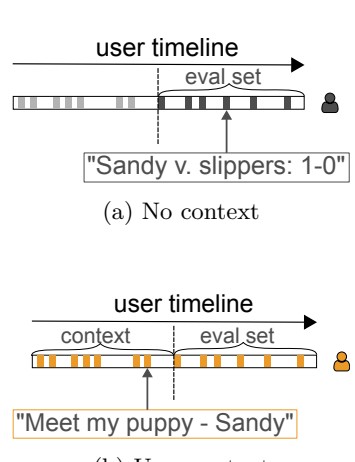

(a) No context

(b) User context

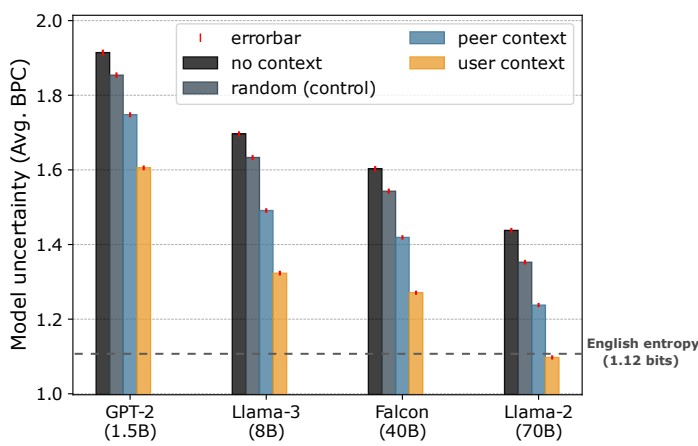

(c) Average bits per character (BPC) required to predict user tweets, with 95% confidence intervals ($N = 5102$). Lower values mean more predictable.

Figure 1: Predictability of a user's tweets using LLMs. Bits per character (BPC) measures, on average, how many bits are required to predict the next character. Predictability improves with additional context to the model: (i) past user tweets (user context, Fig. 1b) (ii) past tweets from the user's peers (peer context) and (iii) past tweets from random users (control). We plot the average BPC over users in Fig. 1c and the estimated entropy rate of the English language from (Takahashi & Tanaka-Ishii, 2018) as comparison. **Most of the predictive information is found in the user context, followed by peer and random context.** Our results are robust across models with different parameter sizes and tokenizers.

et al. (2018). Prediction of sensitive attributes of the user such as age, gender, religion etc. is possible from their peers Jurgens et al. (2017).

Bagrow et al. (2019) go beyond predicting sensitive attributes and look at the predictability of users' online speech. They investigate the theoretical possibility of peer-based user profiling on Twitter. They look at the information content of tweets using a non-parametric estimator, and derive an upper bound on predictability which shows that 8–9 peers suffice to match the predictive information contained in the user's own posts. This upper bound on predictability implies that there *could* exist some predictor which is capable of peer-based user profiling. Our work aims to contextualize their results by empirically testing this hypothesis on concrete predictors. We test this with 15 peers per subject and use transformer-based LLMs, which are currently considered to be the state-of-the-art method for language modeling.

# 3 Experimental setup

We replicate the experimental setup of Bagrow et al. (2019) to measure the predictability of users' online speech using large language models. We define predictability (or rather *un*predictability) as a measure of the model's uncertainty, i.e., its negative log-likelihood, on a specific user's tweets. We observe how model uncertainty changes given *additional* sources of context, specifically:

1. *user context*: past tweets of the user

2. *peer context*: tweets from accounts the user most frequently @-mentions

3. *random context*: tweets from randomly selected users

We start by describing our data collection process (Section 3.1) and what models we used (Section 3.2). Then, we go into detail about the implementation of our prompting and finetuning experiments (in Section 3.3 and 4.3, respectively). We don't share the tweets we collected because of X's Developer Policy. Our code can be found here: `https://anonymous.4open.science/r/twitter-predictability-9BA4`

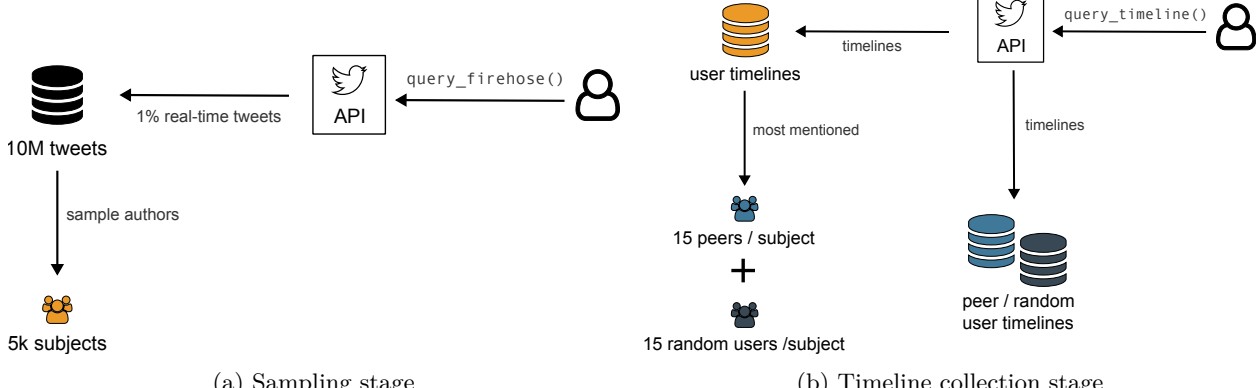

(a) Sampling stage              (b) Timeline collection stage

Figure 2: Our data collection process can be divided into two stages. In the first stage (Fig. 2a), we collected 10M tweets in early 2023 which served as our base for sampling subjects. In the second stage (Fig. 2b), we collected users' timelines.

### 3.1 Data Collection

The data collection process consisted of two main stages: an initial sampling period where we recorded real-time Twitter activity for a month in early 2023 and a second stage where we collected the timelines of sampled users. For a high-level overview of the data collection process, see Figure 2. We collected tweets from three groups of users: (i) *subjects*, approximately 5,000 randomly sampled Twitter users, (ii) *peers* of subjects, which we take to be the top 15 people that each subject most frequently @-mentions, (iii) and *random* users for control purposes. Note that peers defined in this way may include a mix of genuine social contacts, public figures and institutional accounts. We found that 16.7% of peers belong to a verified account [2].

Throughout our data collection, we only collected tweets that were written by the user who posted them (e.g. no retweets) and were classified as English according to Twitter's own classification algorithm. We additionally preprocessed our tweets (removed URLs, special characters, etc.). For more detailed information on the dataset and how it was collected, we refer the reader to Appendix A.1. There we go into detail on what methods we used to achieve a high-quality, representative dataset of English tweets.

#### 3.1.1 Sampling stage

Sampling was done by collecting a pool of tweets whose authors would serve as our base for picking our subjects. We collected them using the Twitter Firehose API; which allowed us to collect a 1% subsample of real-time tweet activity. This collection period lasted roughly 30 days and was done in early 2023 (from 20. January to 10. February), during which we collected **10M tweets** (with ∼5M unique authors). We sampled $N = 5102$ subjects from this pool (0.1% of authors) for our experiments. We filtered out users that scored high (above 0.5 on a scale of 0 to 1) on the Bot-O-Meter bot detection API Sayyadiharikandeh et al. (2020) as well as users that had a high retweet ratio. This ensured the selection of authentic users who had a sufficient amount of self-authored tweets for our dataset.

#### 3.1.2 Timeline collection stage

For each subject, we collected a total of 500 tweets $\mathcal{T}_u = \mathcal{T}_u^{\texttt{eval}} \cup \mathcal{T}_u^{\texttt{user}}$ from their timelines. Half of those tweets were used for estimating predictability $\mathcal{T}_u^{\texttt{eval}}$, while the other half $\mathcal{T}_u^{\texttt{user}}$ served as context. Besides user context, we also introduce peer and random context: $\mathcal{T}_u^{\texttt{peer}}$ and $\mathcal{T}_u^{\texttt{random}}$, again with 250 tweets each. Peer context contained tweets from users that the subject most frequently @-mentioned (top-15). We

---

[2]Only 1.1% had a "blue" verified checkmark (which can be obtained by having a premium subscription). The remaining 15.6% were legacy verified accounts, e.g. news organizations, governments, public figures, etc.

made sure that all context tweets were authored *before* the oldest tweet in $\mathcal{T}_u^{\texttt{eval}}$. In total, we collected approximately **6.25M tweets** from users' timelines.

Similarly to Bagrow et al. (2019), we created a second control group (*temporal control*). However, because of the similarity of the results, we only report results on the random control group (*social control*) in the main text. For results including the temporal control, please see Appendix B.5.

## 3.2 Models

We used four different model families for our experiments, namely **GPT-2** Radford et al. (2019), **Falcon** Almazrouei et al. (2023) as well as **Llama-2** Touvron et al. (2023) and **Llama-3** Dubey et al. (2024). These represent models that share a similar transformer-based architecture that have recently become popular due to their impressive generative capabilities. However, they differ in number of parameters (ranging between 1.5B-70B), training corpus and tokenizers. We further differentiate between base and finetuned versions of these models:

1. *base*: pre-trained LLMs, *no* instruction-tuning (e.g. GPT-2-XL)

2. *tweet-tuned*: base models finetuned on the 10M tweets collected during the sampling stage (e.g. GPT-2-XL-**tt**)

For more details on tweet-tuning, please see Appendix A.2. We considered using GPT-3 and GPT-4, however the OpenAI API unfortunately does not allow access to log probabilities for all tokens (which is necessary for our analysis), only to the top-5. We used Huggingface's transformers library Wolf et al. (2020) to load the models and run our experiments.

## 3.3 Prompting Experiments

Our first approach to measuring model uncertainty is through *prompting*; that is, experiments where we feed a tweet to a model and observe the associated probabilities of outputting that exact tweet. We use **negative log-likelihood (NLL)** as a measure of model uncertainty, and introduce bits per character (or BPC) to enable comparisons across models. The NLL of a tweet $T = (t_1, t_2, ...t_m)$ is commonly defined as follows: $L(T) = -\sum_{t_i \in T} \ln p_\theta(t_i | t_{<i})$, where we use a language model with parameters $\theta$ to predict token $t_i$ based on the preceding tokens $t_{<i}$. Let $\bar{L}_u$ be the average uncertainty associated with predicting tweets of user $u$:

$$\bar{L}_u = \frac{1}{n} \sum_{T_j \in \mathcal{T}_u^{\texttt{eval}}} L(T_j),$$

where $n$ is the total number of tokens. A model-agnostic version of this uncertainty is **bits per character (BPC)**, or otherwise known as *bits per byte*:

$$\overline{bpc}_u = \bar{L}_u \cdot \frac{1}{\bar{C}_u} \cdot \frac{1}{\ln 2}.$$

where $\bar{C}_u$ is the average number of characters per token for user $u$. $\overline{bpc}_u$ tells us the average number of bits required to predict the next character of user $u$'s tweets. In the main text we will commonly report the average BPC over all users, which is $\overline{bpc} = \frac{1}{|\mathcal{U}|} \sum_{u \in \mathcal{U}} \overline{bpc}_u$. Reported results are calculated on $\mathcal{T}_u^{\texttt{eval}}$.

We additionally introduce notation to distinguish what context was used to calculate: $\overline{bpc}^c$, where subscript $c \in \{\texttt{user}, \texttt{peer}, \texttt{random}\}$. Here, the conditional probability of token $t_i$ is based on preceding tokens $t_{<i}$ as well as tokens from the appropriate context: $p_\theta(t_i | \mathcal{T}_u^c, t_{<i})$. The added context lends a convenient cross-entropy like interpretation of the shared information between the context and evaluation tweets we are trying to predict.

We also quantify the average effect size of different contexts on predictability relative to each other. We do this by calculating the **standardized mean difference (SMD)** of the negative log-likelihoods calculated

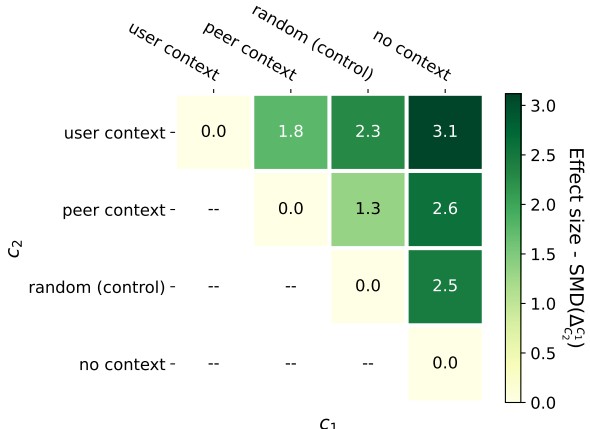

Figure 3: Average effect size of context $c_2$ relative to $c_1$ on model uncertainty. Darker green means greater improvement in model uncertainty (the model becomes less uncertain). For example, user context significantly improves model uncertainty by $3.1\sigma$ over having no context (top right corner). Model: Llama-2-70B.

using different sources of context, i.e. the SMD of $\Delta_{c2}^{c1}(u) = \bar{L}_u^{c1} - \bar{L}_u^{c2}$. Aggregating over all users, this gives us a unit-free quantity to estimate the difference in model uncertainty one context offers over another. Following established practice Cohen (1988), we characterize effect sizes up to $0.2\sigma$ to be small, up to $0.8\sigma$ to be medium and anything above that to be a large effect size. Larger models ($>1.5$B parameters) were loaded using 8-bit precision, which has little to no impact on performance Dettmers et al. (2022). For more details on why we use BPC, sensitivity to prompting strategy, context size, etc. please refer to Appendix A.3.

### 3.4 Finetuning Experiments

In this section we present a second method for estimating model uncertainty, which has three advantages over our previous prompting approach. First, we can fit the entire context into the model compared to prompting where the maximum size of the input sequence is a limitation. Second, it bypasses a common ailment of LLMs: their sensitivity to prompting strategy. Finally, it avoids any limitations in-context learning might have.

Instead of including said context in our prompt, we finetune our model on the different types of context we previously introduced and quantify model uncertainty using its *cross-entropy* loss in the final round of finetuning. We also include experiments on finetuning on a mixture of contexts (e.g. `peer+random` containing tweets from both $\mathcal{T}_u^{\texttt{peer}}$ and $\mathcal{T}_u^{\texttt{random}}$). We combine them by sampling an equal amount of tweets uniformly from each context, while keeping the total number of tweets the same (250 tweets) to enable a fair comparison. We finetuned for 5 epochs with constant learning rate $1e-5$ and batch size 1. We tracked the cross-entropy loss on $\mathcal{T}_u^{\texttt{eval}}$ periodically. Reported results are calculated on $\mathcal{T}_u^{\texttt{eval}}$.

## 4 Results

In our experiments, we quantify how additional context influences model uncertainty, i.e., predictability. We used two methods of feeding additional information to our model, with complementary strengths and weaknesses: one prompting based approach which evaluates in-context learning and one finetuning based approach. We start with evaluating how in-context learning performs on *base models* (Section 4.1), and what they learn from the additional context (Section 4.2). Next, we evaluate *tweet-tuned* models, where we replicate our main findings by finetuning on different types of contexts (Section 4.3). We also show that it is surprisingly hard to predict user tweets using large language models, even with additional context

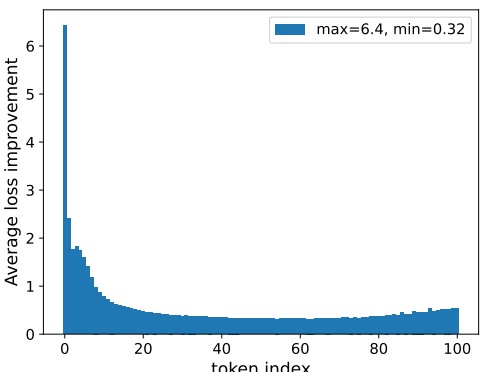

Figure 4: Average improvement in NLL from additional user context (compared to none). The first few tokens of a tweet benefit most from the additional context. Model: Llama-2-70B.

(Section 4.4). Finally, we provide some insight into how our results extend to different subgroups within our population (Section 4.5).

## 4.1 Added Context Reduces Uncertainty in Base Models

We start with the results of the prompting approach on base models. In Figure 1, we plot the average BPC calculated over all of our subjects for four different models with varying parameter sizes. The first token was left out of the analysis (see Appendix B.6 for more detail). The no context case was consistently the most unpredictable with the highest BPC. Depending on what context was included, the amount of improvement varied. We show that most of the predictive information is found in the user context, followed by peer and random context. This trend is consistent across all of our models. For users that are hard to predict, we observe slightly larger improvements in model uncertainty (App. B.4).

We also quantify the effect size each context had on predictability. We plot this for each context pair in Figure 3 on Llama-2. This effect size matrix illustrates our finding that all types of context have a large effect size ($> 2.5\sigma$) over having no context at all (last column). It also allows us to quantify the effect size of our main finding: user context offers $1.8\sigma$ improvement over peer context — an even bigger improvement than what peer context offers over random context ($1.3\sigma$). While these effect sizes are most pronounced on Llama-2, we found that these results also translate to our other models (Figure 21 in Appendix).

## 4.2 Base Models Learn Syntax, Among Other Things

Next, we were curious where and how context improves predictability. Interestingly, even *random* context improves predictability in a non-negligible way. We found evidence of the model learning to use the right *syntax*; i.e. how long tweets are, the presence of @-mentions and hashtags, etc. Indeed, the predictability of the '@' token improved the most compared to all other tokens (App. B.7) when using random context.

Predicting @-mentions and hashtags correctly is also a contributing factor in the case of user and peer context. Visualizing the improvement in predictability over individual tokens inside a tweet we noticed an interesting phenomenon (Figure 4). Locations of greatest improvement were typically at the start and at the end of tweets – where @-mentions and hashtags would often be located. Assuming these were indeed one of the greatest sources of error in the no context case, removing them would make our tweets more predictable overall, and the effect of additional context on predictability less pronounced. Indeed, in Figure 5 subjects became more predictable on average, and the effect of context on predictability decreased (by $\sim 20\%$ for random context, $\sim 4\%$ for peer context and $\sim 7\%$ for user context on Llama-2 70b). In other words, models learned to assign higher probability to frequently used @-mentions and hashtags from context. This variation partly reflects differences in syntax composition across context types: random context contains $\sim 35\%$ more

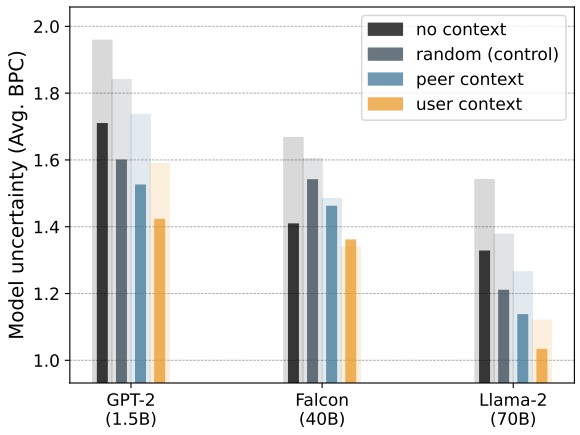

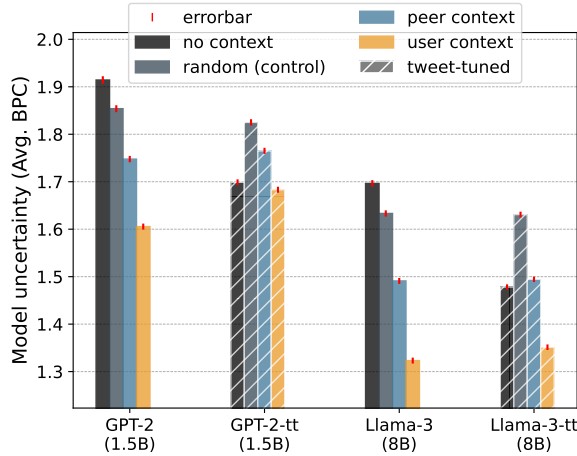

Figure 5: Average model uncertainty on tweets without @-mentions and hashtags. Subjects become more predictable on average, and the positive effect of context on predictability decreases. Lighter bars are the results from our original experiment for comparison.

Figure 6: Average model uncertainty on base vs. tweet-tuned models. After tweet-tuning, in-context learning on additional context offers little to no improvement.

hashtags per tweet than peer context (0.27 vs. 0.20), while user context contains $\sim 29\%$ more @-mentions per tweet than peer context (0.99 vs. 0.77). The remaining 80–96% of the contextual gain is not explained by syntax; what exactly is learned beyond syntax remains an open question. While in some cases context ceases to be useful, the relative comparisons between different types of contexts remain the same.

### 4.3 Results on Tweet-Tuned Models

Next, we prompt on tweet-tuned models. We plot the model uncertainty before and after tweet tuning for GPT-2-XL and Llama-3-8B in Figure 6. For both, tweet-tuning lowers model uncertainty in the no context case. However, predictability does not improve anymore with additional context in most cases. This is in line with our observations from before: base models learn typical twitter syntax from context. Our tweet-tuned models learn this through finetuning, limiting the amount of useful information that can be leveraged from in-context learning. For GPT-2-XL-tt, we find limited to no improvement through additional context; only Llama-3-8B-tt is able to learn more from additional user context. To overcome these limitations we observed in in-context learning, we instead turn to finetuning tweet tuned models on the context.

Figure 7a shows the average loss curves (computed over 1000 subjects) for finetuning GPT-2-XL-tt on different types of contexts [3]. Again, for all three contexts the loss goes down significantly. However, the final loss they converge to is different, with large gaps between each. If we order contexts based on the achieved loss in the final round, we get the same order as before: user context is best, followed by peer context, then random context. Combined with the observed stability of user rankings across models (App. B.1), we believe this result would extend to larger models as well.

We established that user context always outperforms peer context, regardless of model choice or experimental method. Still, one might argue that by *combining* user and peer context, one might achieve better results. This would suggest that there is some non-overlapping predictive information inside the peer context wrt. the user context. We test this hypothesis by finetuning on a mixture of contexts (Fig. 7b). Combining `peer+random` contexts resulted in a linear interpolation of the final losses of finetuning on either context. In other words, mixing random and peer context outperformed the final loss of finetuning exclusively on random context by a large margin. Here, we found evidence of additional predictive information in the peer context,

---

[3]A clarifying note on the "jumps" in the loss curves (at steps 15, 20, ... etc.): Since each $\mathcal{T}_u^c$ contains 250 tweets of varying lengths, 5 epochs of training resulted in different global steps for each user-context combination.

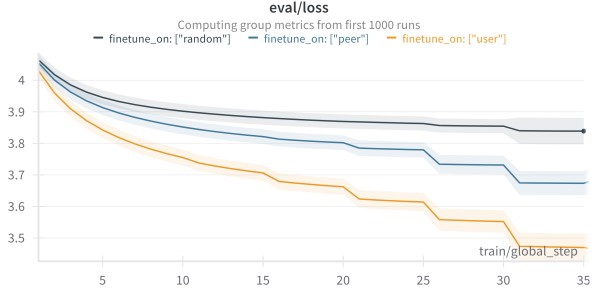 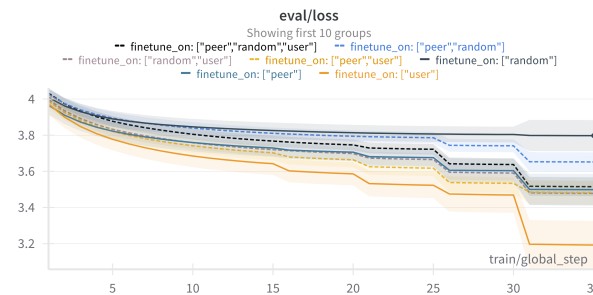

(a) Finetuning on user, peer and random context exclusively. Again, most predictive information can be found in the user context, followed by peer then random context.

(b) Finetuning on mixtures of contexts. Mixing contexts does not always lead to better results, suggesting overlap in predictive information.

Figure 7: Average loss curves with standard error for finetuning experiments on GPT-2-XL-tt. For each subject ($N = 5102$) we finetune on the specified context, and compute the cross-entropy loss on $\mathcal{T}_u^{\texttt{eval}}$. The plotted averages are computed over the loss curves of 1000 subjects.

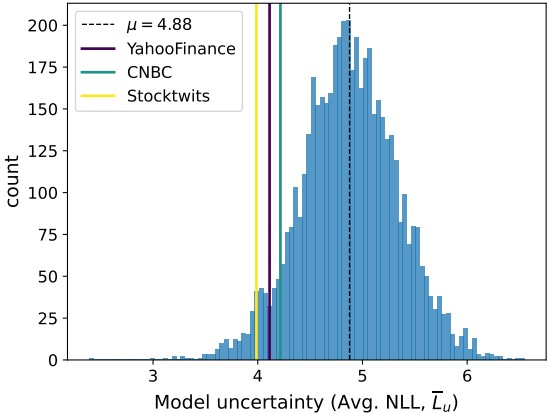

Figure 8: Distribution of the average NLL of our subjects (with no additional context). As comparison, we include the average NLL of three financial news accounts: YahooFinance, CNBC and Stocktwits. Model: GPT-2-XL.

which was not contained in the random context. However, mixing `peer+user` contexts did *not* result in a significantly lower final loss. This suggests a significant overlap in predictive information between peer and user context, upholding our claim that user context is strictly better than peer context.

## 4.4 User Tweets are Hard to Predict

In Figure 8 we present the distribution of the average model uncertainty on the tweets of our subjects. To gain an intuition on how hard it is to predict our subjects' tweets relative to other accounts, we included three popular financial news accounts as comparison: YahooFinance, CNBC, and Stocktwits. Intuitively, predicting financial news and the stock market is hard Johnson et al. (2003), which should make them less predictable. However, Figure 8 reveals that most subjects in our pool are actually *harder* to predict than those news accounts. Of course, while the *content* of those tweets may be hard to predict, their *style* and *vocabulary* may not.

Finally, the absolute information content of users' tweets is between 1.5 and 2 bits per character, depending on which model we use (see 'no context' bars in Figure 1c). As a comparison, Shannon's upper bound on the

cross-entropy of the English language is 1.3 bits per character Shannon (1951), and recent estimates using neural language models say it is as low as 1.12 bits Takahashi & Tanaka-Ishii (2018). These results suggest that individual pieces of our online expression taken out of context are far from predictable using today's LLMs. Provided with additional user context, only our largest model (Llama-2 70b) achieves comparable entropy, with an average of 1.1193 bits per character. While humans converge to 1.3 bits after only seeing ~32 characters Moradi et al. (1998), language models typically need hundreds to thousands of tokens to converge (see Fig. 23 in Appendix B.9).

### 4.5   Results across groups

It is also important to analyze our results and see how they apply to different demographics. Are some groups more predictable than others? Do we discover the same relative relationships between contexts? For example, it may be possible that members of some group may be more likely to be profiled through their peers. While we did not have any demographic information that covered all our subjects, we came up with proxies that allowed us to analyze a subset of them. We analyzed our results by gender, location, and their intersection. The latter provided similar results as the first two and can be found in Appendix B.10.

We extracted 830 users' preferred pronouns as a proxy for gender. This was done by searching for string matches in their profile descriptions. For example, profiles with "he/him" in their description were matched to 'masculine', "she/her" to 'feminine', and ones that contained a mix of both or other pronouns [4] were matched to the 'diverse' category. What we mean by a 'proxy' is that it merely implies the individual's preferred linguistic gender, not necessarily their social gender Devinney et al. (2022) (i.e. she/her can be the preferred pronoun of both women and trans women). Results from Figure 9a show that LLMs achieve similar performance across all categories. Subjects across different groups seem to be equally predictable. Even the relative relationship between contexts is preserved — no group is significantly more likely to be profiled through their social circle.

We further identified 2221 users' country based on the specified location in their profile. This was mostly done by using widely available geocoding services like Nominatim [5]. We only analyzed the top-5 most common countries (United States, Great Britain, Canada, India and Nigeria). Compared to before, results in Figure 9b again show similar performance across countries except for Nigeria, where model uncertainty is significantly higher. This indicates that language models perform significantly worse on tweets belonging to Nigerian profiles. This may be in part due to Nigerian subjects speaking their own English dialects. Our results closely mirror prior work which has found that models perform significantly worse on certain dialects, such as African-American-English Blodgett et al. (2016).

## 5   Limitations

**Negative-log likelihood as a measure of uncertainty**   While negative log-likelihood is the most popular measure of model uncertainty, its main limitation is that it does not take into account any improvements on semantically equivalent text Kuhn et al. (2023). Another related issue is calibration Xiao et al. (2022). We also found the models to be highly sensitive to slight changes in prompting. See the Appendix for how the separator token between tweets A.3, the first token B.6, and tweet length B.3 affect model uncertainty. While these points might affect absolute numbers in our analysis, they do not affect our main statements about the relative comparisons between different models and contexts. Despite these limitations, NLL is the standard metric optimized during language model training and a widely used proxy for downstream performance Saunshi et al. (2020). While downstream performance does not imply concrete risks such as impersonation, we avoid testing for them due to ethical concerns.

**Data contamination**   Furthermore, we cannot rule out the possibility that some tweets in our corpus were part of the training dataset of the LLMs that we used. Contamination may result in lower model uncertainty. However, we believe the risk and severity of data contamination to be limited; we provide more detail as to why in Appendix A.2.

---

[4] The full list of pronouns we matched for is in Appendix B.10.

[5] https://nominatim.org/

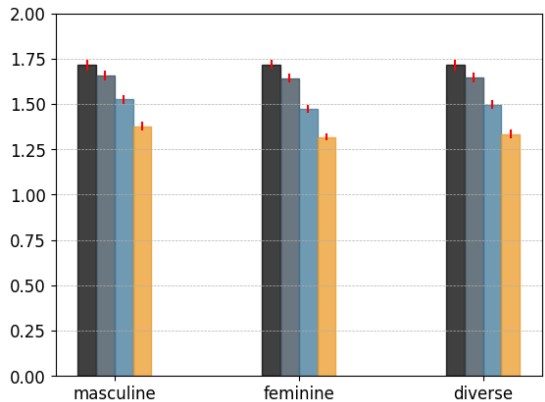 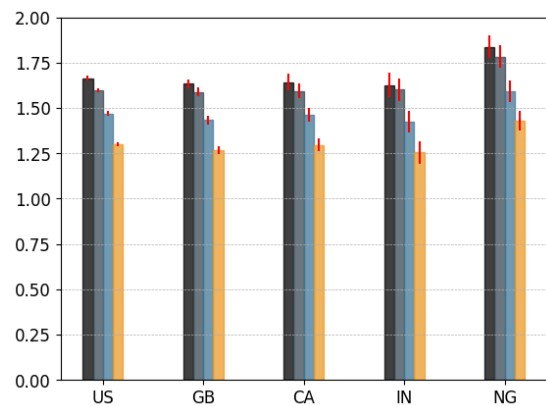

(a) Results by gender. 'masculine': he/him (233), 'feminine': she/her (291), and 'diverse' (306): mix of feminine / masculine or other pronouns (like they/them).

(b) Results by location. Top-5 locations: 'US': United States (1555), 'GB': Great Britain (408), 'CA': Canada (124), 'IN': India (70) and 'NG': Nigeria (64).

Figure 9: Results across different demographic groups. Model: Llama-3-8B.

**Subject sampling**   Our subjects are sampled from users active during a 30-day collection window, which may bias the sample toward more active users — a limitation we share with Bagrow et al. (2019), whose sampling methodology we follow. Importantly, the 30-day window applies only to subject *identification*: the subsequent timeline collection retrieves each subject's full tweet history (see Appendix A.1).

**Peer definition**   Following Bagrow et al. (2019), we define peers as the accounts a user most frequently @-mentions. Alternative definitions could be based on follows, likes or retweets. Users typically follow hundreds to thousands of accounts (median: 302), which makes it difficult to filter for true connections. Compared to likes and retweets, @-mentions (which include replies) are more likely to reflect meaningful interactions with the mentioned account and their content. As a result, peer sets based on @-mentions includes a mix of genuine social contacts, public figures and institutional accounts: 16.7% of peers belong to a verified account. Whether alternative peer definitions would change our findings remains an open question for future work.

**External validity**   Our analysis is constrained to the English-speaking population. How our results extend to other languages and dialects would be an extremely valuable avenue for future work. While Twitter is not the only widely used social media platform, it is one of the primary text-focused ones. Our findings may extend to other similar platforms (such as Mastodon, BlueSky and Threads), but they may apply less to platforms that focus more on sharing image/video content (such as Instagram and TikTok). Furthermore, our analysis considers text in isolation; in practice, social media posts are often reactions to linked content, images, or ongoing interactions, which are not captured by our text-based measure of predictability.

## 6   Discussion

We presented the results of an investigation using four large language models into the predictability of online speech by analyzing posts on X (Twitter). As the basis of our study, we collected posts from more than five thousand users' timelines and their peers. We used a total of 6.25M tweets for our main experiments, plus an additional 10M for tweet-tuning.

Our main finding is that online speech remains hard to predict with the large language models we study. Most users' tweets are less predictable than tweets about financial news, and even with additional context, predictability remains relatively low. Only our largest model with additional user context is able to approximate the estimated entropy rate of the English language. We also found that additional context improves the prediction of basic signals: specifically, guessing the correct hashtags and @-mentions. All in all, our

findings suggest that despite the impressive capabilities of large language models in other areas, the models we study predict speech rather poorly.

Similarly, our results indicate that it matters greatly what context we use for prediction. We questioned whether the predictive information inside peer tweets is enough to match (or even surpass) those of the user's own tweets, as suggested by Bagrow et al. (2019). We believe that this is unlikely. Our results show that user context consistently outperforms peer context in a manner that is robust to model choice and evaluation method. Additional experiments suggest that whatever predictive information is inside the peer context, can also be found in the user context.

To summarize, we have shown that predicting online speech is rather difficult and that it depends a lot on what information is used as context. Our findings are robust across methods (prompting vs. finetuning) and show consistent trends across different demographic groups (gender and location).

## Broader Impact

Our study contributes empirical evidence to debates about LLM-enabled privacy risks on social media. We find that individual online speech remains difficult to predict with the models we evaluate, which informs questions about risks such as shadow profiling or impersonation. Whether the predictability levels we measure are sufficient or insufficient for such downstream harms is an open question; our results establish a baseline but cannot rule these risks out. We do observe substantial variability across users (Appendix B.2), suggesting that individual-level risks may differ considerably from population-level averages. Our demographic analyses rely on imperfect proxies (self-reported pronouns, profile location fields), and group-level findings should be interpreted with appropriate caution. Moreover, profiling risks may also arise from non-linguistic signals — such as behavioral patterns, social graph structure, metadata, or images — which fall entirely outside the scope of our text-based analysis. Future work could attempt to build a bridge between model uncertainty and downstream capabilities that affect the individual, ensuring continued scrutiny as language models inevitably evolve.

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

## A  Experimental Setup

### A.1  Data collection

#### A.1.1  Sampling stage

We used the Twitter Firehose API to query a 1% sub-sample of real-time tweet activity in early 2023 (from 20. January to 10. February). We used the following filter expression to query the API:

```
'sample:1 followers_count:0 -is:retweet lang:en'
```

Where the options have the following meaning:

- `sample:1`: Return a 1% sub-sample of the filtered tweets. (The specified number has to be between 1-100 representing a % value.)

- `followers_count:0`: Return tweets made by users with at least 0 number of followers. This is a dummy filter because sample/is/lang are not standalone filters (filters that can be used on their own) and need additional standalone filters (like followers_count) to work.

- `-is:retweet`: Don't return retweets.

- `lang:en`: Return English tweets.

This collection phase resulted in a pool of 10M tweets, with 5M unique authors. We sampled our subjects from this pool of authors ($N = 5102$ sample). Strictly speaking, our sample will be biased towards users that have 1) been more active during our initial collection period and 2) are more active users in general. This was the closest we could get to a random sample with the offered API endpoints. This practice follows Bagrow et al. (2019)'s method of randomly sampling users.

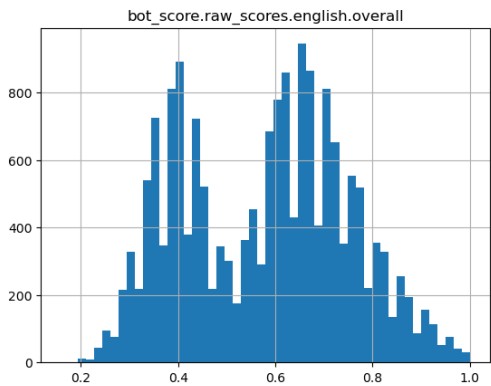

Figure 10: Distribution of Bot-O-Meter scores. We only selected users in our subject pool that had a score that was lower than 0.5.

Twitter had roughly ∼500M active monthly users in 2023 Yaccarino (2023) – some of which may have been bot accounts Lee et al. (2011). To address this issue, we decided to filter out accounts that scored high (above 0.5 on a scale of 0 to 1) on the Bot-O-Meter bot detection API Sayyadiharikandeh et al. (2020). Bot-O-Meter classifies users on a scale from 0-1 based on 200 of their tweets. See Figure 10 for a distribution of these scores. We dropped users that had a score higher than 0.5. Additionally, we dropped users that had a high retweet ratio (more than 80% of their tweets consisted of retweets). This is an additional measure to prevent bot accounts in our subject pool (bots are known for frequently retweeting content Yang et al. (2020); Gilani et al. (2017)) as well as a practical consideration since we only wanted to include non-retweets in our analysis.

### A.1.2  Timeline collection stage

We used the Twitter API's Timeline endpoint to query the subjects' most recent tweets. To get tweets from around the same timeframe, we choose an `end_time` (which was the start of our sampling stage, 20. January). Tweets made after `end_time` were not included. Per subject, we collected a total of 500 tweets, half of which was reserved for evaluation, the other half served as user context. Some user tweets date back as early as 2011, however the vast majority (95%) were authored after 2022.

From these tweets, we identified the top-15 most mentioned users (the user's peers) and collected 50 tweets from their timelines as well. These, as well as the user context tweets were collected such that they were authored before the oldest evaluation tweet ($t^*$ in Figure 11). From these, we selected the 250 most recent tweets. A simpler approach would have been to select exactly $\frac{250}{15} \approx 17$ tweets per peer. However, not all peers had this many tweets on their timeline before $t^*$, hence our choice for collecting more tweets per peer than necessary.

In the end, we had queried the timelines of around ∼90k users, with 15M timeline tweets in our database. We sampled from this pool of users and tweets for our social / temporal control. The social control consists of tweets of 15 random users with $\frac{250}{15} \approx 17$ tweets each. We made sure that the sampled user did not coincide with the subject itself / any of their peers. The temporal control on the other hand contains tweets that were made around the same time as the tweets inside peer control. Again, we made sure that the tweets' authors did not overlap with the peer pool or the subject. Figure 11 illustrates all of our settings, while Figure 12 shows the time histogram of an example dataset belonging to one of our subjects.

### A.1.3  Preprocessing tweets

Before our experiments, we preprocessed our collected tweets by filtering out urls, deduplicated spaces and fixed some special character encodings. We found that urls were not relevant in analyzing the predictability of (organic) online speech, while deduplication of spaces is a fairly common preprocessing step in NLP.

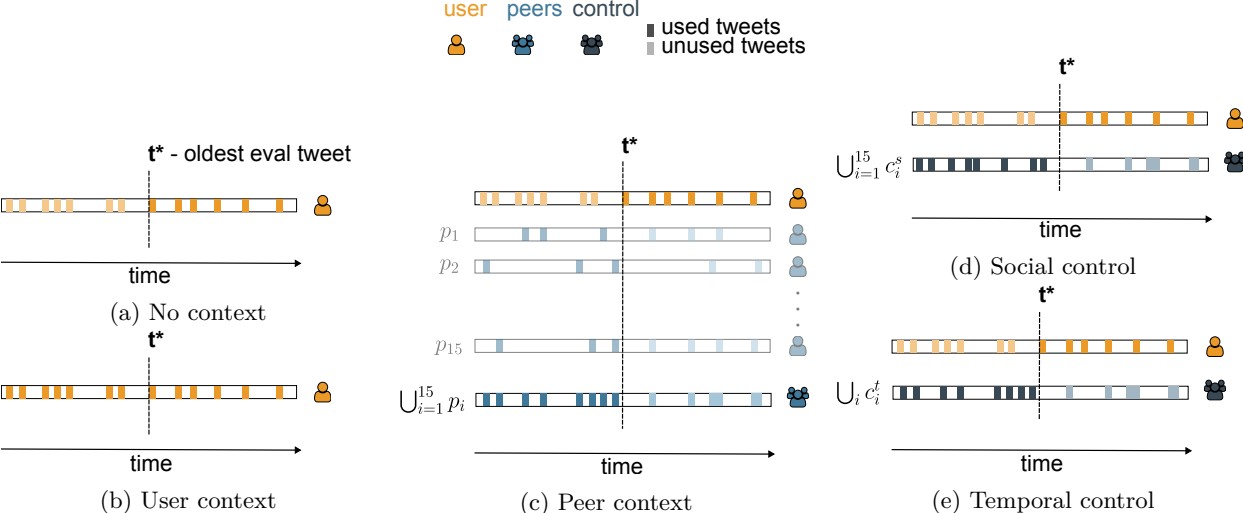

Figure 11: Data creation protocol. We evaluate predictability on a set of evaluation tweets, and how it changes depending on what context we provide.

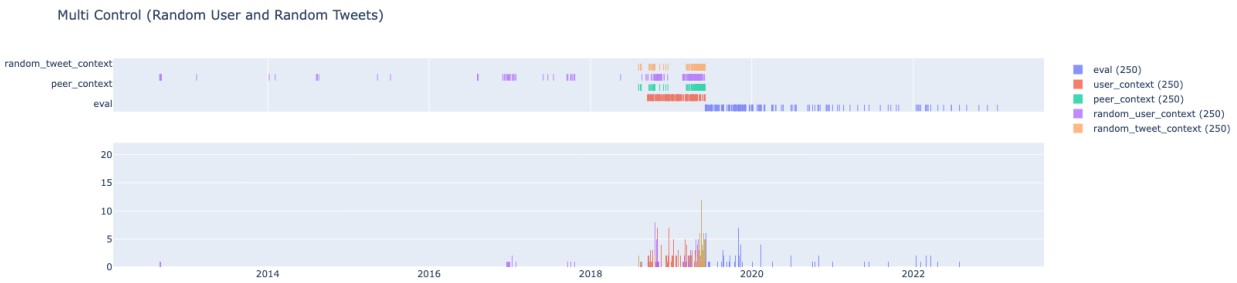

Figure 12: Time histogram of an example dataset of some subject $u$. We illustrate $\mathcal{T}_u^{\texttt{eval}}$, and how all context tweets were written before the oldest tweet in $\mathcal{T}_u^{\texttt{eval}}$. The social control contains tweets from random users, while the temporal control contains tweets that were authored around the same time as the peer tweets.

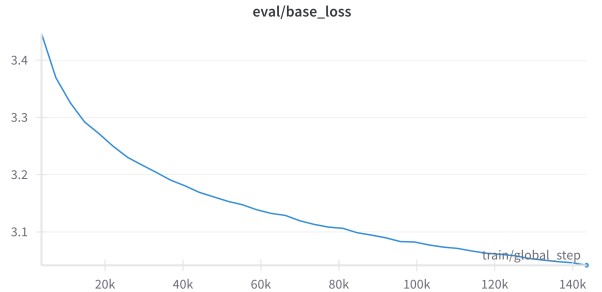
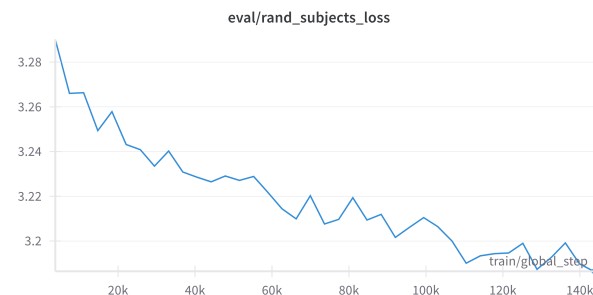

(a) Loss calculated on the 5% validation split ($\sim$ 0.5M tweets).

(b) Loss calculated on the evaluation set of 100 subjects (25k tweets). There is a lot more stochasticity, which in part is due to the smaller sample size and a less diverse pool of authors.

Figure 13: Loss curves of tweet-tuning GPT-2-XL on 10M tweets for 1 epoch. Validation loss (negative log-likelihood) was calculated on a 5% split. To make sure that tweet-tuning also improved subject predictability, we also selected a random subset ($n = 100$) of the 5k subjects. We used their evaluation tweets to calculate the loss and since each $|\mathcal{T}_u^{\texttt{eval}}| = 250$, this means a total of 25000 tweets.

These preprocessing choices were partially inspired by an open-source project called HuggingTweets Dayama (2022). For our experiments described in Section 4.2, we further filtered out @-mentions and hashtags.

## A.2 Models

### A.2.1 Training data

GPT-2 was trained on WebText with content up to December 2017 Radford et al. (2019) (95% of our subjects' tweets were authored after 2022), Falcon was trained on RefinedWeb which is is built using all CommonCrawl dumps until the 2023-06 one Penedo et al. (2023) (CommonCrawl typically does not contain snapshots of Twitter com (2024)) and the authors of the Llama models claim to have "made an effort to remove data from certain sites known to contain a high volume of personal information about private individuals." Touvron et al. (2023); Dubey et al. (2024). Llama-2 has a knowledge cutoff at September 2022, while Llama-3 8B has seen data up to March 2023. Based on these facts we believe there is limited risk of data contamination.

### A.2.2 Tweet-tuning on 10M tweets

We finetuned some models on the 10M tweets we collected during the sampling stage of our data collection process (as described in Section 3.1), where a 5% split was reserved for validation. Tweets were concatenated, with the special eos token '<|endoftext|>' serving as a separator between them.

**GPT-2-XL-tt**   We finetuned all parameters for 1 epoch, using a constant learning rate of 5e−5, batch size 8 and `fp16` mixed precision training on a single A-100 80GB GPU with an AdamW optimizer. We used the example finetuning script from the transformers library as our base[6], where we kept most of the default training arguments.

With a batch size of 8, it took 147279 global steps to go over the entire set of training tweets once. In addition to the evaluation loss (NLL on the 5% validation split, which was checked periodically) we also tracked the loss calculated on the combined evaluation set of 100 subjects (25000 tweets in total) to make sure that the pre-finetuning improved prediction on our subjects as well. We present a figure of the loss curves in Figure 13.

---

[6]`run_clm.py` from here: `https://github.com/huggingface/transformers/tree/main/examples/pytorch/language-modeling`

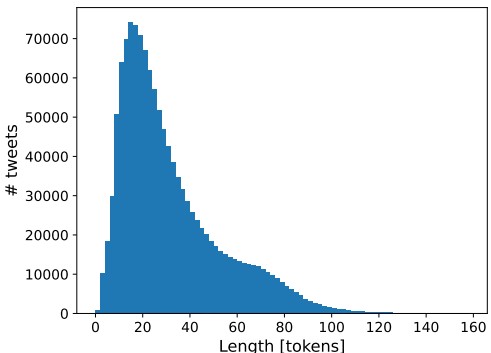

Figure 14: Distribution of tweet length in tokens (Llama-2 tokenizer). Tweets are from $\mathcal{T}^{\text{eval}}$

**Llama-3-8b-tt**   We finetuned all parameters for 5 epochs, using an initial learning rate of $2\mathrm{e}{-5}$ and a cosine learning rate scheduler. We used a paged AdamW optimizer with batch size 1 and gradient accumulation steps set to 8. We used axolotl for finetuning, the base for our config was their example full finetune script[7] for Llama-3-8b (only the initial learning rate and number of epochs were changed).

## A.3   Prompting experiments

Our first approach to measuring model uncertainty is through *prompting*; that is, experiments where we feed a tweet to a model and observe the associated probabilities of outputting that exact tweet. It is important to note that this approach does **not** involve content generation. This allows us to avoid a host of additional modeling choices, making prompting a more robust method. We use negative log-likelihood (or NLL) as a measure of model uncertainty, and introduce bits per character (or BPC) to enable comparisons across models. Reported results are calculated on $\mathcal{T}_u^{\text{eval}}$.

Let us denote a single tweet as $T = (t_1, t_2, ...t_m)$, where $t_i$ is a single token ($i = 0...m$). A token is an item in the LLM's vocabulary, and can be thought of as a collection of characters that frequently co-occur. Each tweet has a maximum of 280 characters[8] and after tokenization most tweets have between 0-100 tokens (Fig. 19). We use a language model with parameters $\theta$ to predict token $t_i$ based on the preceding tokens $t_{<i}$. The model's output will be a likelihood over all possible tokens in the model's vocabulary, however we are only interested in the conditional probability of $t_i$: $p_\theta(t_i|t_{<i})$.

**Calculating negative log-likelihood**   Our metric for predictability is the average negative log-likelihood (or NLL for short from now on). We define the NLL of a tweet $T$ in the following way: $L(T) = -\sum_{t_i \in T} \ln p_\theta(t_i|t_{<i})$. This gives us an estimate of the model's uncertainty when predicting the tokens inside tweet $T$. The average uncertainty over all tweets in the evaluation set $\mathcal{T}_u^{\text{eval}}$ is

$$\bar{L}_u = \frac{1}{n} \sum_{T_j \in \mathcal{T}_u^{\text{eval}}} L(T_j),$$

where $n$ is the total number of tokens. We additionally introduce notation to distinguish what context was used to calculate: $\bar{L}_u^c$, where subscript $c \in \{\texttt{user}, \texttt{peer}, \texttt{random}\}$. Here, the conditional probability of token $t_i$ is based on preceding tokens $t_{<i}$ as well as tokens from the appropriate context: $p_\theta(t_i|\mathcal{T}_u^c, t_{<i})$. The added context lends a convenient cross-entropy like interpretation of the shared information between the context and evaluation tweets we are trying to predict.

**Sensitivity to prompting strategy**   Tweets that served as context were concatenated using the 'newline' token for GPT-2-XL and the 'space' token for the other models (they had no standalone 'newline' token like

---

[7]https://github.com/axolotl-ai-cloud/axolotl/blob/main/examples/llama-3/fft-8b.yaml
[8]This limit changed to 4000 characters on the 09. February 2023.

GPT-2). Using low-frequency tokens as a separator between tweets (such as the eos token, which is usually reserved for separating training documents) produced abnormally high NLLs, which is why we decided to use more common tokens, such as space or newlines. We used the following input sequence lengths: 1024 tokens for GPT-2-XL, 2048 for Falcon 40B and 4096 for Llama-2 and Llama-3 to calculate the token probabilities. In case the provided context exceeded this length, the oldest tweets were discarded.

**Conversion to bits per character**  Models with different tokenizers produce NLLs that are not commensurate with each other due to a different set of tokens being used. To overcome this, we convert our measure to a metric called *bits per character*, also known as *bits per byte*. Let $c_t$ be the number of characters in token $t$. We define the number of characters inside tweet $T$ as $C(T) = \sum_{t_i \in T} c_{t_i}$. Taking the average over all tokens in $\mathcal{T}_u$ we get $\bar{C}_u = \frac{1}{n} \sum_{T_j \in \mathcal{T}_u} C(T_j)$. We define bits per character (BPC) formally as

$$\overline{bpc}_u = \bar{L}_u \cdot \frac{1}{\bar{C}_u} \cdot \frac{1}{\ln 2} \, .$$

This number tells us the average number of bits required to predict the next character in the set $\mathcal{T}_u^{\texttt{eval}}$. This leads to a more interpretable variant of the common perplexity measure [9]. The average BPC over all users is $\overline{bpc} = \frac{1}{|\mathcal{U}|} \sum_{u \in \mathcal{U}} \overline{bpc}_u$. Similarly to before, we define $\overline{bpc}^c$ as the measure of model uncertainty where token probabilities were calculated including tokens from some context $c$.

**Estimating the effect size of context on predictability**  Next, we are interested in measuring the average effect size of different contexts on predictability. At a user level, we quantify the difference in predictability between context $c1$ and $c2$ as $\Delta_{c2}^{c1}(u) = \bar{L}_u^{c1} - \bar{L}_u^{c2}$. Aggregating over all users, we define the standardized mean difference (SMD) of $\Delta_{c2}^{c1}$ for each context pair:

$$\texttt{SMD}(\Delta_{c2}^{c1}) = \frac{\mu_{\Delta_{c2}^{c1}}}{\sigma_{\Delta_{c2}^{c1}}}$$

This gives us a unit-free quantity to estimate the effect size. Following established practice Cohen (1988), we characterize effect sizes up to $0.2\sigma$ to be small, up to $0.8\sigma$ to be medium and anything above that to be a large effect size.

### A.4  Finetuning experiments

Tweet-tuned models were *additionally finetuned* on one of the contexts. We also experimented with using *mixtures* of different contexts. Here, our proxy for unpredictability was the negative log-likelihood (or also commonly known as the *cross-entropy* loss) in the final round of fine-tuning. Tweets were concatenated, with the special eos token '<|endoftext|>' serving as a separator between them. Reported results are calculated on $\mathcal{T}_u^{\texttt{eval}}$.

We only ran this experiment on the tweet-tuned version of GPT-2 (*GPT-2-XL-tt*), because of the non-negligible amount of time and resources this experiment requires to run. Per subject there are 3 different types of context, plus 4 mixtures of contexts. That is, we finetuned GPT-2 a total of $7 * 5102 = 35714$ times. Together with tweet-tuning, it cost us approximately 1.6 e19 FLOPs to run this experiment. [10]  While finetuning on larger models would certainly result in lower cross-entropy overall, we do not believe that it would change our main conclusions. This is supported by the high agreement on the ranking of users across models (Fig. 15) as well as the stability of our conclusions for our prompting experiments as we go up in model size.

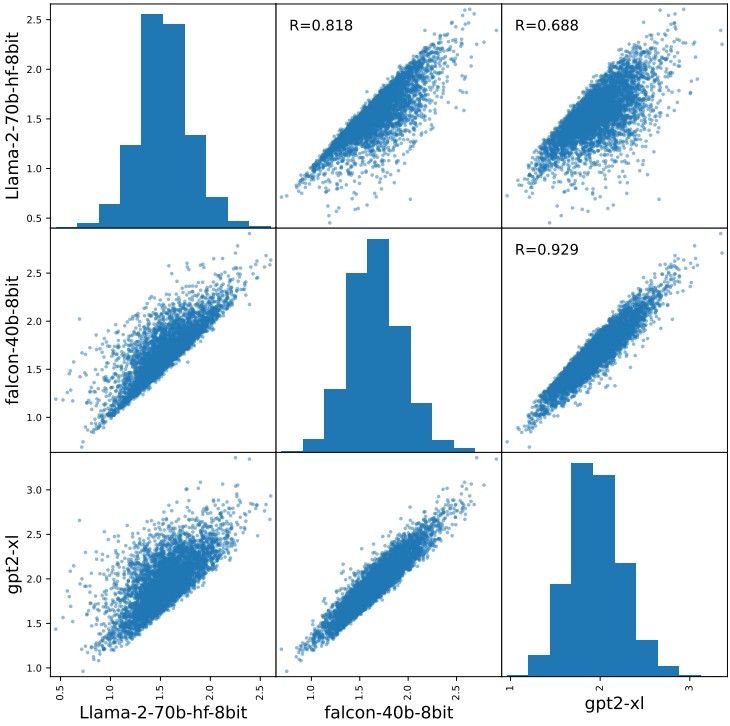

Figure 15: There is high agreement on the ranking of users based on predictability across models. We measure predictability in the no-context setting ($\overline{bpc}_u$; bits per character) for each user. A high-scoring user (who is harder to predict) will get a similarly high score on a different model (conversely a low-scoring user will get a lower score).

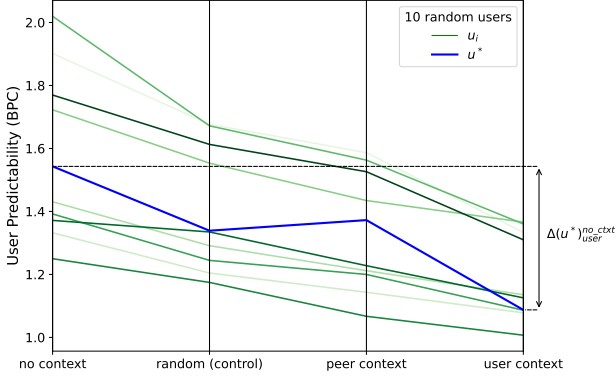

Figure 16: Differences in predictability $\Delta_{c_2}^{c_1}$ using the Llama-2 model. We picked 10 random users and plot their predictability using different contexts (y-axis). Comparing across contexts we get the difference in predictability: $\Delta(u_i)_{c_2}^{c_1}$. While model uncertainty goes down on average as we evaluate on more predictive contexts, there can be individual level variance that does not necessarily always follow this trend.

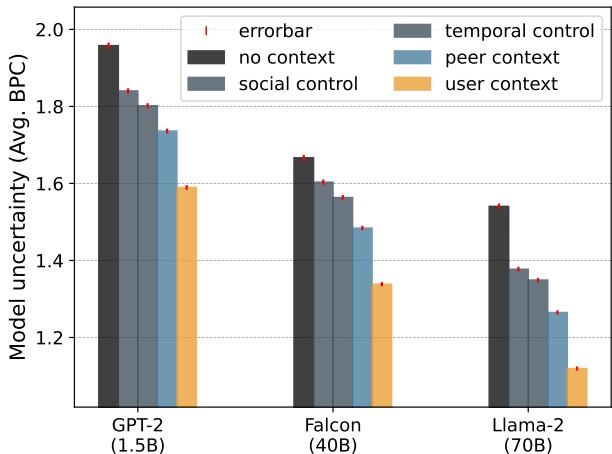

Figure 17: Average bits per character (BPC) required to predict user tweets, with 95% confidence intervals. Here both control settings are included: (i) past tweets from random users (social control, left) and (ii) past tweets made around the same time as the peer tweets (temporal control, right). These results are consistent with the ones presented in Fig. 1.

# B   Supplementary Results

## B.1   High agreement on user ranking

We find that the ranking of users according to average unpredictability of their tweets is strongly correlated between all three of our models (Figure 15). This means that a user who scores high (i.e. their tweets are harder to predict) according to one model, will likely score high on a different model as well. This points to a certain robustness of the negative log-likelihoods: Although the absolute numbers change from one model to the next, the ranking of users is similar.

## B.2   Individual variability

While globally there is a tendency where user context outperforms peer context, peer context outperforms random context, etc., there is substantial variability on the individual level as illustrated in Figure 16. In the highlighted blue example, random context improves predictability more than the peer context.

## B.3   Correlation with average tweet length

In Figure 14 we show the distribution of subjects' tweet length in tokens. Most of the tweets have between 0 and 100 tokens. We also present the relationship between model uncertainty and the average tweet length of a given subject in Figure 19. NLL and average tweet length are correlated: users with longer tweets are on average more predictable.

## B.4   Hard to predict users gain slightly more from additional context

In Figure 15 we have shown that if we rank users according to how predictable they are, there is a high agreement across models wrt. this ranking. Now, with additional context, we analyze how the change in predictability is influenced by this ranking (Figure 20). We find that with additional context, improvement in predictability is larger for users who are already hard to predict. Predictability improves on average by $\sim 0.1$ bits for every 1 bit increase in "difficulty to predict".

---

[9]Perplexity is the exponentiated average negative log-likelihood.

[10]As a comparison: it took OpenAI around e20-e21 FLOPs to finetune GPT-2 *from scratch*.

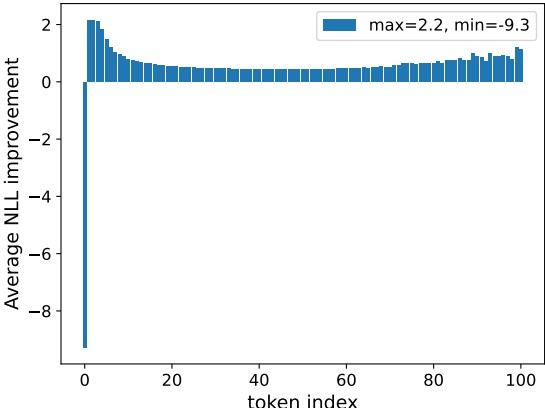

Figure 18: Average improvement in NLL from additional user context (compared to none). The deterioration of predicting the first token correctly is an artifact of how we ran our experiments. Model: Llama-3-8B.

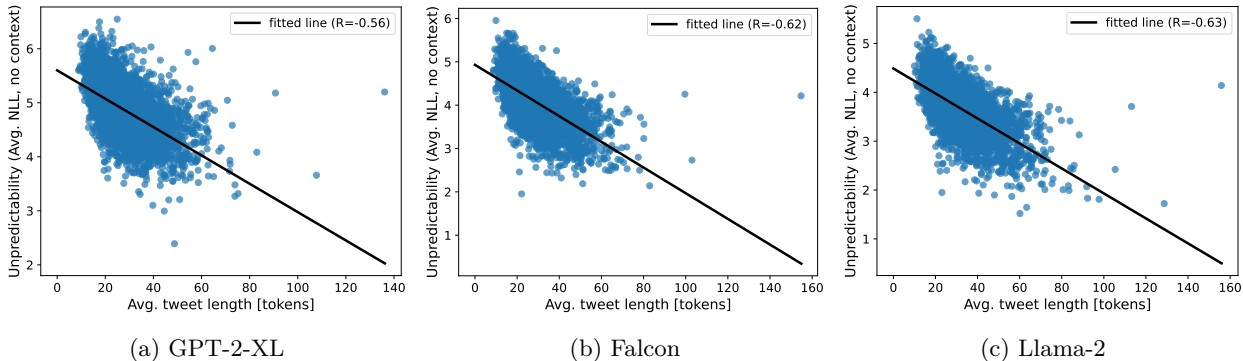

(a) GPT-2-XL            (b) Falcon            (c) Llama-2

Figure 19: Users with longer tweets are more predictable. On the y-axis, we plot the average NLL required to predict a user's tweets (no context setting). The x-axis shows the user's average tweet length (in tokens).

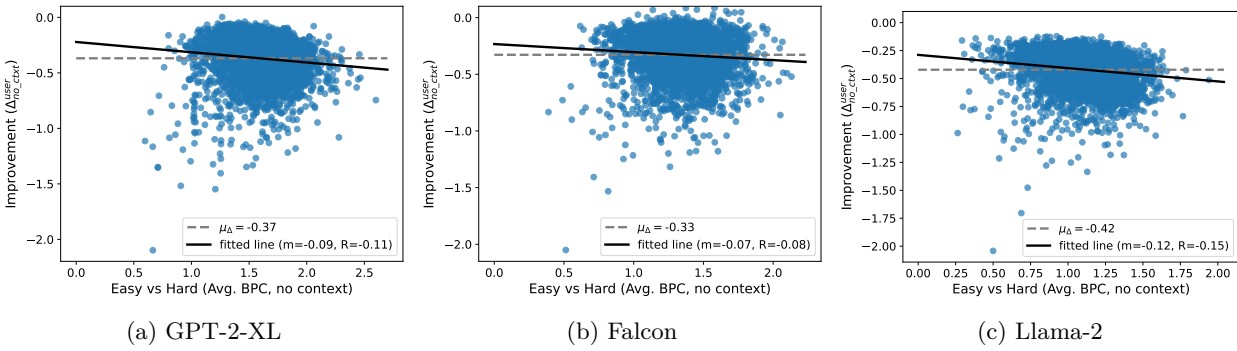

(a) GPT-2-XL            (b) Falcon            (c) Llama-2

Figure 20: Hard to predict users get more predictable with additional context, but only slightly. On the x-axis, we plot the average BPC required to predict a user's tweets (no context setting), effectively sorting them based on how predictable they are (easy vs. hard to predict). The y-axis plots the relative improvement (decrease in BPC) with additional user context.

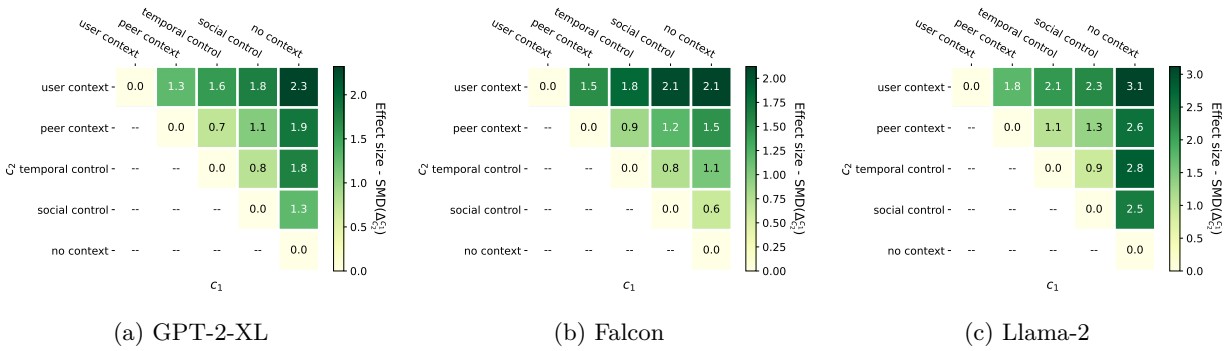

(a) GPT-2-XL  (b) Falcon  (c) Llama-2

Figure 21: Average effect size of $c_2$ relative to $c_1$ on user predictability. We look at the difference in predictability for each user $\Delta_{c2}^{c1}(u) = \bar{L}_u^{c1} - \bar{L}_u^{c2}$, where $\bar{L}_u^c$ is the average negative log-likelihood of user $u$ under context $c$. Plotted values are standardized mean differences (SMD) of $\Delta_{c2}^{c1}$. Darker green means greater improvement.

### B.5 Two control groups: social and temporal control

Bagrow et al. (2019) introduce two control groups in their experiment: a *social* and a *temporal* control. Social control includes tweets from 15 random users. Temporal control on the other hand, selects 250 tweets that were authored around the same time as the tweets from the peer context. See Figure 11 for an illustration of both. The rug plot on the top of Figure 12 shows the same, but on real data of a random subject. Figure 17 shows our main results including both controls, while Figure 21 shows the corresponding effect sizes.

### B.6 First token uncertainty

Adding context can help in correctly predicting the first token. This is the case for GPT-2, Falcon and Llama-2 models (see Fig. 4), where models learn from context that a tweet often begins with an @-mention. For Llama-3, we see a slightly different picture in Figure 18: predicting the first token correctly becomes *harder* because of the added context. Upon investigating this difference, we found that this is simply an artifact of how we ran our experiments. Llama-3 assigns much higher probability to predicting '@' as the first token to begin with. With the added context however, it assigns high probability to ' @' (a space followed by an @), which is encoded as a single token in Llama-3 ([571]). Because of how we ran our experiments, the space ([31]) and the subsequent @-mention ([220]) got encoded separately, resulting in tokens [31, 220] instead of the expected [571]. Thus, we decided to exclude the first token from our analysis when reporting results on Llama-3.

Figure 22 illustrates how dropping the first token from our analysis affects model uncertainty across all four models. Unsurprisingly, model uncertainty goes down. For models which had high negative log likelihoods on the first token improved the most in the no context setting. On the other hand – for reasons we outlined above – model uncertainty on the Llama-3 model improved the most *outside* the no context setting. Overall, by excluding the first token from our analysis, we found that our main results from Figure 1 are nicely replicated on Llama-3 as well.

### B.7 Most improved token due to random context: @-mention

We were interested in which tokens benefited most after including random (i.e. social) context. We selected tokens with >100 occurrences, and ranked them based on how much their prediction improved on average. Table 1 shows that the '@' token benefits the most across all three models, indicating that @-mentions are locations of greatest improvement. See Table 2 for a full table on top-10 most improved token predictions (with all possible contexts).

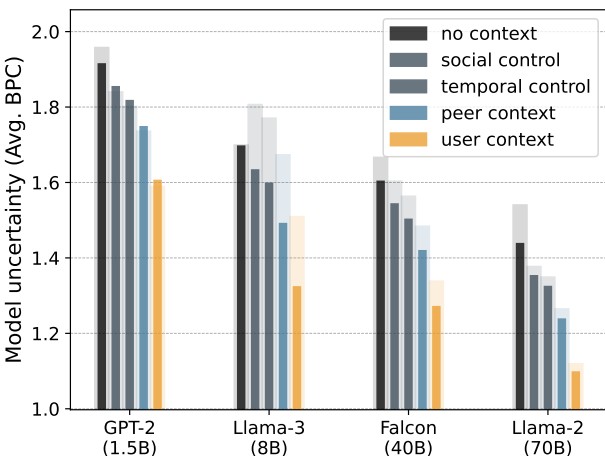

Figure 22: The effect of taking the first token out of the analysis. Overall, the average model uncertainty decreases (predictability goes up). Lighter bars are results from our original experiment.

|    | **GPT-2-XL** | **Falcon** | **Llama-2** |
|----|----|----|----|
|    | social control | | |
| 1  | @ | @ | _@ |
| 2  | âĠ | Brit | _Hey |
| 3  | Ļ | ĠAmen | _Ain |
| 4  | ĠARTICLE | ĠNah | _Happy |
| 5  | ĠðŁij | ĠBru | _Wait |
| 6  | ĠðŁ | ĠSame | _tf |
| 7  | ĠâĠ | ĠDamn | /@ |
| 8  | Ġâĺ | ĠWait | rach |
| 9  | Ġâľ | ĠDang | _Okay |
| 10 | ľ | ĠNope | _Ton |

Table 1: Top ten tokens whose predictability went up the most (on average) after including tweets from random users as context (compared to no context). Notice how the @ sign is the token that got "bumped" the most, suggesting that the additional random context helped with predicting @-mentions. (Ġ is a special symbol for the space character in the GPT-2-XL and Falcon tokenizers.)

### B.8  Falcon-40b w/o @-mentions and hashtags

We repeated our prompting experiments after removing @-mentions and hashtags from our tweets (see Figure 5). An interesting observation for our results on Falcon is that after removing @-mentions and hashtags, additional context did not improve predictability the same way as it did before. In case of peer and random context, it even increased model uncertainty. The effect size of the user context dropped to only $0.4\sigma$, which is significantly less than the $2.1\sigma$ from before. A possible explanation is that the only (useful) predictive signal Falcon was able to pick up on was in the removed pieces of text. Another possibility is that it is simply more sensitive to discontinuities inside the text than other models.

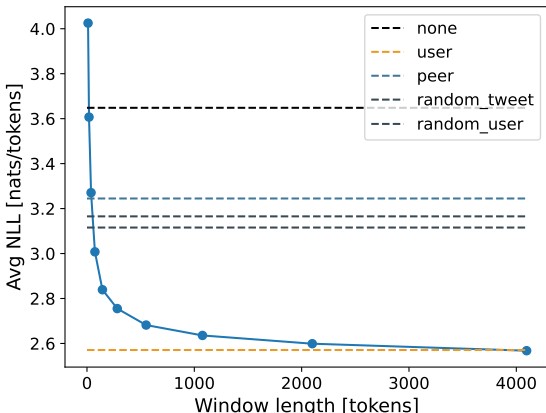

Figure 23: Model uncertainty on a random subject. Increasing the context window size lowers model uncertainty. Dashed lines are model uncertainties reached with 4096 context window size, with the specified context. Model: Llama-2-70b.

### B.9  Negative log-likelihood convergence

We illustrate how negative log-likelihood converges as we increase the context window size. In other words, we show that the more tokens (tweets) we include, the lower NLL we get. We concatenated tweets in $\mathcal{T}_u^{\texttt{eval}}$ for a random user $u$ using the 'space' token as a separator. In Figure 23, we show how NLL converges as we increase the context window size from 10 tokens to 4096 tokens. In the end, it converges to the same NLL we reach when we condition on concatenated tweets from the user context $\mathcal{T}_u^{\texttt{user}}$. As we can see, LLMs may need a fairly long context size for the negative log-likelihood to converge.

### B.10  Results across groups

### B.10.1  Gender

We applied the following regular expressions on users' names, descriptions and location to match their preferred pronouns. We used the 'feminine', 'masculine' and 'diverse' categories. Case was ignored.

The regex for the 'feminine' category:

```
(\b(?:she|her|hers|herself)(?:\s*[\s/|]\s*(?:she|her|hers|herself))?\b)
```

The regex for the 'masculine' category:

```
(\b(?:he|him|his|himself)(?:\s*[\s/|]\s*(?:he|him|his|himself))?\b)
```

The regex for the 'diverse' category:

```
(\b(?:they|them|their|theirs|themself|themselves|ze|zir|zirs|zirself|fae|faer|faers|faerself \
```

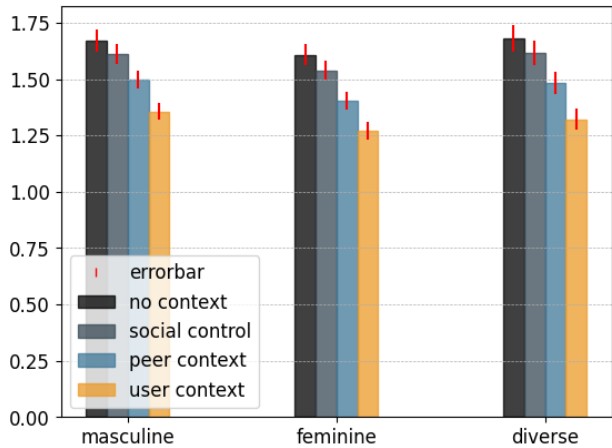

Figure 24: Intersection of gender and US. Categories: US & masculine (78), US & feminine (69) and US & diverse (81). Model: Llama-3-8B

```
|xe|xem|xyr|xyrs|xyrself|ey|em|eir|eirs|eirself|ve|ver|vis|verself|per|pers|perself)(?:\s*[\s \
/|]\s*(?:they|them|their|theirs|themself|themselves|ze|zir|zirs|zirself|fae|faer|faers|faerself \
|xe|xem|xyr|xyrs|xyrself|ey|em|eir|eirs|eirself|ve|ver|vis|verself|per|pers|perself))?\b)
```

Subjects that matched for pronouns in both the 'feminine' and 'masculine' category were also put in the 'diverse' group.

### B.10.2 Location

Users' location was determined using geolocating services like Nominatim. We used python libraries like `geopy` and `geotext` to extract the country code of the location. Another method was to parse flag emojis and convert them to the corresponding two-letter country code. The input was the specified location in users' profiles.

### B.10.3 Intersection of gender and US

We also had enough data to analyze the intersection of US subjects and the gender category. Results are in Fig. 24. We see similar predictability and relative relationships in these subcategories as well.

| | **GPT-2-XL** | | | |
| | social | temporal | peer | user |
|---|---|---|---|---|
| 1 | @ | @ | hetti | ĠKraft |
| 2 | âĠ | âĠ | DOM | hetti |
| 3 | Ļ | Ļ | 204 | ĠARTICLE |
| 4 | ĠARTICLE | ĠARTICLE | Agg | hyde |
| 5 | ĠðŁij | ĠðŁij | rium | DOM |
| 6 | ĠðŁ | ĠâĠ | alys | ĠSag |
| 7 | ĠâĠ | ĠðŁ | perm | medium |
| 8 | Ġâĺ | ĠPis | Hour | ĠRescue |
| 9 | Ġâİ | İ | hyde | ĠPis |
| 10 | İ | Ġâĺ | Extra | ĠKeller |

| | **Falcon** | | | |
| | social | temporal | peer | user |
|---|---|---|---|---|
| 1 | @ | @ | hyde | 749 |
| 2 | Brit | Brit | perm | Ġgenealogy |
| 3 | ĠAmen | ĠNah | Agg | Ich |
| 4 | ĠNah | ĠBru | 018 | ildo |
| 5 | ĠBru | ĠAmen | 641 | hyde |
| 6 | ĠSame | ĠDamn | atts | MCA |
| 7 | ĠDamn | ĠWait | 576 | perm |
| 8 | ĠWait | ĠSame | 931 | ĠKeller |
| 9 | ĠDang | ĠDude | Chi | ENV |
| 10 | ĠNope | ellan | 454 | ĠVenue |

| | **Llama-2** | | | |
| | social | temporal | peer | user |
|---|---|---|---|---|
| 1 | __@ | __@ | Extra | __Kraft |
| 2 | __Hey | __Ain | DOM | medium |
| 3 | __Ain | __Hey | member | Extra |
| 4 | __Happy | __tf | zent | zent |
| 5 | __Wait | Fil | $v$ | $v$ |
| 6 | __tf | __Wait | Los | __Via |
| 7 | /@ | soft | members | hour |
| 8 | rach | rach | __@ | member |
| 9 | __Okay | __Via | gat | DOM |
| 10 | __Ton | __Om | __txt | ihe |

Table 2: Top ten tokens whose predictability went up the most (on average) after including tweets from some context (compared to no context). Tokens with $> 100$ occurrences were selected.

