# OpenReview forum: "Limits to Predicting Online Speech Using Large Language Models"
_TMLR — Under review for TMLR_

### Review · Reviewer_gHy8 · 2026-04-14

**Summary Of Contributions:**

This paper studies how predictable individual users’ online speech is on X using LLMs. The authors collect 10M tweets for domain adaptation and 6.25M timeline tweets from 5,102 users together with tweets from their peers and random-control accounts, and evaluate predictability using negative log-likelihood / bits-per-character under different context conditions. Across four LLMs, they find that predictability depends strongly on context: a user’s own history is most informative, peer context is less informative, and random context is least informative among the non-empty context conditions. Beyond this main comparison, the paper examines which information is being exploited in context, finding that syntax-like markers such as @-mentions and hashtags account for part of the gain, and it supplements the prompting results with tweet-tuning, per-user finetuning, and subgroup analyses.

**Additional Comments:**

Minor nitpicks:
- In multiple places in the document, \citet style citations were used instead of \citep. Please correct that
- Please pass the text through a grammar checking software; there were some issues with the writeup.

**Audience:**

Yes

**Audience Explanation:**

I think the study is interesting to some communities. Naturally, NLP is a clear fit. However, the work might also be relevant to privacy and security research, and potentially to AI Fairness, Ethics, and Bias as well.

**Broader Impact Concerns:**

I could not find a broader impact statement in the paper. The introduction and the final paragraph of the discussions section do acknowledge privacy- and profiling-related concerns. However, I would recommend thinking a more explicit, broader-impact discussion clarifying that the reported predictability results do not by themselves rule out downstream harms such as profiling, impersonation, or targeted manipulation, and a more careful discussion of the limitations of the demographic proxy analyses.

**Claims And Evidence:**

Yes

**Claims Explanation:**

I will go through the claims that I think are supported.

**Predictability depends strongly on context (user context > peer context > random context > no context):** I think this is the paper's strongest claim and is well supported by the evidence. Figure 1 shows the same ordering across all four models in the main prompting experiments. Figure 3 shows that user context provides a larger gain ($1.8\sigma$) than peer context does ($1.3\sigma$) over random context. Figure 21 further shows that this qualitative pattern holds across models, even though the magnitudes vary, with Llama-2 showing the strongest separation.
- **All forms of added context help relative to no context.** I also think the experiments support this weaker but still important claim. Across the main prompting results, all three context conditions improve predictability relative to the no-context baseline, and the effect-size analysis in Figure 3 shows large improvements for each of them.

**Random context provides a real, nontrivial gain:** I think the observation that the random-control condition is consistently better than no context across models is interesting. It suggests that some part of the gain comes from generic platform-level regularities rather than user or peer-specific information. The authors connect this to syntax, but I think it is still an interesting observation that, at least to me, was not trivial.

**Peer context contains predictive information beyond random context:** Figure 7b seems to show that peer context contains a useful signal beyond what is present in the random control, since peer+random improves over random alone. Despite my concerns about the peer definition (see requested changes) I think this narrower conclusion follows from the reported experiments.

**Syntax explains the contextual improvement:** I think this is supported in a limited sense. Figure 5 shows that removing @-mentions and hashtags reduces part of the contextual gain, and Section 4.2 quantifies that reduction. So I do think the paper supports the claim that syntax-like markers explain some of the improvement. However, I also think this result needs reframing, since the strongest numerical effect appears in the random-control condition rather than in the more socially relevant peer and user settings. See the requested changes.

**Online speech is (surprisingly) hard to predict:** In the no-context setting, prediction is at roughly 1.5 to 2 bits per character across models, and that only the largest model with user context approaches the cited English entropy estimate of 1.12 bits. This supports the narrower claim that held-out user tweets remain difficult to predict for current LLMs under the paper’s protocol. But I would be careful in the framing here (see requested changes section). At the moment,I think this result tells more about the model-data interactions, rather than the data itself.

**Predictive information inside peer context can also be found in user context:** The evidence suggests overlap, but the stronger conclusion seems to rely too heavily on one specific finetuning experiment (GPT-2-XL-tt).  So I would treat this as suggestive, but not fully established.

**Requested Changes:**

**Syntax.** The “up to 20%” figure applies only to the random control condition, whereas the more important peer and user contexts show much smaller reductions of roughly 4% and 7%, respectively. It seems to me that the largest effect comes from the least socially meaningful setting, but it motivates a broader mechanistic story. Moreover, for the peer and user conditions that matter most to the paper’s claims, the analysis fails to explain the improvement. The current ablation therefore supports only a limited claim that @-mentions and hashtags account for a small part of the gain, not the stronger implication that syntactic learning is a central explanation of the result.

**Definition of peer:** In Section 2, peers are defined as the top 15 accounts that each subject @-mentions most frequently. I am not fully convinced that this is a reliable proxy for socially meaningful peers. Users could also @-mention customer bots, news outlets, celebrities, not just actual friends or close contacts. This matters for the experiments: If the peer set contains a substantial number of institutional or public accounts, then the weaker performance of the peer context may partly reflect a noisy proxy rather than a genuine limit of peer information. Such accounts could bring in extra heterogeneity in style, topic, and function, which could make the peer condition harder to use even if true peers do contain predictive signal. This could help explain why peer context is consistently better than random context but still substantially worse than user context in both the prompting and finetuning results. Finally, It can weaken the interpretation of the mixture experiment in Section 4.3: if the peer condition is already diluted, then the fact that peer+user does not improve much over user alone is less informative, because it may simply indicate that the chosen peer proxy is weak.

I think the authors should first establish which account types are actually selected as peers in the dataset. If this proxy is mostly clean, the issue should still be explicitly discussed as a limitation. If it is not, then an ablation over alternative peer definitions or filtering strategies would be important.

**Prior knowledge in the model:** More generally, I am not fully convinced by the framing around the intrinsic unpredictability of online speech. It is not clear to me that the reported unpredictability is a property of the data alone, rather than a joint property of the data distribution and the model’s prior knowledge. Since these models are heavily pretrained, and the paper cannot fully rule out some degree of overlap, a model with different prior exposure to social-media text or different domain adaptation could plausibly produce different uncertainty values without any change in the underlying tweets. To me it seems that the paper is really measuring the residual uncertainty of online speech for already-pretrained models, not the intrinsic unpredictability of online speech in general. I think this distinction should be made more explicit in the discussion and conclusion. I would be happy to discuss this further with the authors.

---

> ### Author Response · Authors · 2026-05-07
> **Author Response**
>
> Thank you for your careful and constructive reading. We appreciate your detailed assessment of which claims are well-supported — particularly the context ordering and the non-trivial gain from random context. We hope the revisions below adequately address your concerns about framing.
>
> ### Syntax
>
> > "The 'up to 20%' figure applies only to the random control condition; peer and user contexts show smaller reductions of ~4% and ~7%."
>
> We agree and have revised all passages to highlight this variation accurately. In Results, Section 4.2 this was already explicitly specified: "~20% for random context, ~4% for peer context and ~7% for user context". Upon inspection, we found that the differences in data composition could explain the *direction* of this variation: random context contains ~35% more hashtags/tweet than peer context, and user context contains ~28% more @-mentions/tweet than peer context (see table below) — contexts with more hashtags/@-mentions show a higher syntax effect. We now include this observation in Section 4.2 in our revision.
>
> | Context split | Avg. hashtags/tweet | Avg. @-mentions/tweet |
> |---|---|---|
> | `user_context` | 0.197 | 0.987 |
> | `peer_context` | 0.195 | 0.769 |
> | `random_context` | 0.267 | 0.790 |
>
> However, the composition difference does not fully explain the *magnitude* of the variation. A potential explanation is that user and peer contexts provide richer non-syntactic signal - user-specific vocabulary, topics, and writing style — which dilutes the relative contribution of syntax to the total contextual gain; but this explanation remains speculative as it is not directly tested in our experiments.
>
> > "@-mentions and hashtags account for a small part of the gain, not a central explanation of the result."
>
> Thank you for making this important distinction. We agree, and have revised all passages that implied more importance to syntax than what the numbers suggest. We maintain that improvements from random context are significantly influenced by syntax, but we now use more careful phrasing when extending it to peer and user context.
>
> ---
>
> ### Peer definition
>
> > "I am not fully convinced that this is a reliable proxy for socially meaningful peers. Users could also @-mention bots, news outlets, or celebrities."
>
> We understand the reviewer's concern, and we have run an analysis to quantify the amount of peers that fall outside the friend / social contact circle. We found that 16.7% of selected peers hold a verified account, confirming that peer sets include public figures and institutional accounts. We concede that this was not clear before, and have updated parts of the paper where 'peer' and 'social contact' were used interchangeably. It is hard to say whether this explains the gap between peer and user context. After all, socially meaningful contacts are not the only ones influencing us: news accounts, celebrities or the aptly named "influencers" play an important role too. To summarize, we agree that our definition of peers does not fully reflect social ties, but it's unclear whether this affects our results.
>
> > "If this proxy is mostly clean, the issue should be discussed as a limitation; if not, an ablation over alternative peer definitions would be important."
>
> In the Limitations section, we added a paragraph that discusses alternative definitions based on follows, likes, and retweets. However, we have not used them for the following reasons: users typically follow hundreds to thousands of accounts (median: 302), making it difficult to isolate true connections; likes and retweets are equally sparse signals and require less engagement than @-mentions (which include replies), making them a noisier proxy of meaningful interaction. Another reason to keep this definition is to remain consistent with Bagrow et al. (2019), whose  methodology we follow.
>
> ---
>
> ### Prior knowledge framing
>
> > "I think this result tells more about the model-data interactions, rather than the data itself."
>
> We fully agree and have revised the paper throughout: the Introduction now reads "LLMs struggle to predict online speech" and the Discussion "online speech remains hard to predict with the large language models we study." This positions uncertainty as residual uncertainty of the model, rather than the intrinsic unpredictability of online speech.
>
> ---
>
> ### Broader impact
>
> > "I could not find a broader impact statement in the paper."
>
> We have added a new Broader Impact section acknowledging that our results inform but do not resolve questions about shadow profiling and impersonation, that individual-level risks may differ from population-level averages (pointing to the individual variability appendix), and that profiling risks arising from non-linguistic signals fall outside our conclusions. The final paragraph of the Discussion has been removed and its content moved to the new Broader Impact section.

---

### Review · Reviewer_JZ2s · 2026-04-15

**Summary Of Contributions:**

This paper investigates the extent to which large language models can predict an individual’s online expression on X, and whether such prediction benefits more from the user’s own prior posts or from the posts of their social peers. To study this question, the authors construct a large-scale dataset, perform tweet-domain adaptation, and evaluate multiple language models under four context conditions: no context, random context, peer context, and user context. The main finding is consistent across models: user context yields the largest improvement in predictability, followed by peer context, then random context, while individual online expression remains difficult to predict overall. The paper also analyzes the role of social-media-specific tokens such as @-mentions and hashtags, and includes finetuning experiments to support the main conclusions.

**Audience:**

Yes

**Audience Explanation:**

1.The empirical study is  in scale with a large user set, a large tweet corpus, and evaluation across multiple models and context settings.
2.One of the main result is clear and consistent: user context is more informative than peer context, and peer context is more informative than random context.
3.The paper provides an negative result: even strong LLMs do not make individual online expression easily predictable.
4.The paper is well organized, and the limitations are acknowledged with reasonable transparency.

**Claims And Evidence:**

Yes

**Claims Explanation:**

1.The empirical study is  in scale with a large user set, a large tweet corpus, and evaluation across multiple models and context settings.
2.One of the main result is clear and consistent: user context is more informative than peer context, and peer context is more informative than random context.
3.The paper provides an negative result: even strong LLMs do not make individual online expression easily predictable.
4.The paper is well organized, and the limitations are acknowledged with reasonable transparency.

**Requested Changes:**

1. The practical significance of the main conclusion is limited. Since social media posts are often driven by external events, images, videos, links, and ongoing interactions, predicting posts from historical plain text alone makes the reported “unpredictability” less informative in practice.
2. The dataset is large, but its representativeness remains insufficiently characterized. A key concern is not the dataset size, but the subject sampling protocol. Subjects are first identified from a roughly 30-day window of platform activity, which may bias the sample toward users who were more active during that period and more active in general. Although the subsequent timeline collection is broader, the paper does not fully establish how well this sample reflects the stable and general patterns in the platform.
3. Part of the context gain, particularly for random context, appears to be driven by platform-specific surface syntax. Clarifying how much of the remaining gain reflects deeper user-specific or peer-specific information would make the interpretation of the main result more precise.
4. The conclusions are supported across several open models, but their generality to the latest frontier models remains unverified. While the paper evaluates four open models, but excludes latest LLM, and the finetuning experiment is run on models with 8B because of computational cost. This supports the main trend within the tested setup, but not a broad claim of robustness across current LLMs.
5. The evaluation does not consider semantic-level equivalence between textual content. A post may be predictable in meaning even if its exact wording is not. The paper only includes metrics on token-level that may overestimate the claimed unpredictability.
6. The applicability boundary of the core claim remains under-specified relative to the evidence. The study is restricted to English-language posts on X, relies on an activity-dependent sampling pipeline, and evaluates only four open-weight model families. Under this evidence base, the main conclusion should be scoped more carefully, since its generalization to other platforms, user populations, and model settings remains unclear by evidence.

**Minor issues
1. Some missing words and grammatical errors are found in the paper. For example, in Page 2 contribution 2, subject "we" is missing in the sentence "In Section 4.1 show that a user’s own posts have significantly more predictive information than posts from their close social ties."
2. The reference should be carefully revised. Please carefully check the references and replace preprint-only entries with formal published versions wherever available.

---

> ### Author Response · Authors · 2026-05-07
> **Author Response**
>
> Thank you for the thorough and fair review! We appreciate your recognition of the empirical scale, the consistency of the main result across models, and the overall organization of the paper. We hope the revisions below adequately address your concerns.
>
> ### Practical significance
>
> > "The practical significance of the main conclusion is limited. Since social media posts are often driven by external events, images, videos, links, and ongoing interactions, predicting posts from historical plain text alone makes the reported 'unpredictability' less informative in practice."
>
> We agree, and have added a sentence to the External validity paragraph explicitly acknowledging that social media posts are often reactions to linked content, images, or ongoing interactions not captured by our text-based measure. Twitter/X is among the largest text-focused social media platforms, making it the most natural setting for a text-based study; extending to other modalities is an interesting direction but outside the scope of this study.
>
> ---
>
> ### Dataset representativeness
>
> > "Subjects are first identified from a roughly 30-day window of platform activity, which may bias the sample toward users who were more active during that period and more active in general."
>
> True, sampling from a 30-day activity window does bias toward more active users, and we have added a "Subject sampling" paragraph to Limitations acknowledging this. We note that this limitation is shared with Bagrow et al. (2019), whose methodology we follow as the closest approximation to random sampling available via the Twitter API. The 30-day window applies only to subject *identification*; the subsequent timeline collection retrieves each subject's full tweet history.
>
> ---
>
> ### Syntax mechanism
>
> > "Part of the context gain, particularly for random context, appears to be driven by platform-specific surface syntax. Clarifying how much of the remaining gain reflects deeper user-specific or peer-specific information would make the interpretation more precise."
>
> Section 4.2 already specifies that syntax accounts for ~20% of the contextual gain for random context, ~7% for user context, and ~4% for peer context. We have now propagated this context-dependent framing to the Abstract and Introduction, where previously only the ~20% figure appeared. The remaining 80-96% of the contextual gain could potentially be attributed to user-specific or peer-specific information. The specific breakdown of what is learned in-context beyond syntax remains to be determined in future work.
>
> ---
>
> ### Frontier models
>
> > "This supports the main trend within the tested setup, but not a broad claim of robustness across current LLMs."
>
> At the time these experiments were run, the models we used represented the current frontier in open-source models; the landscape has since shifted. However, most frontier models today are not open-source, and their APIs typically do not expose token-level log-probabilities — a strict requirement of our NLL-based method. We have updated all state-of-the-art language to scope conclusions to the models we evaluate, and the final sentence of the Broader Impact section explicitly acknowledges that conclusions may need to be revisited as models evolve.
>
> ---
>
> ### Semantic equivalence
>
> > "A post may be predictable in meaning even if its exact wording is not. The paper only includes metrics on token-level that may overestimate the claimed unpredictability."
>
> Our study follows the information-theoretic tradition of Bagrow et al. (2019) where NLL/BPC is the principled measure; semantic equivalence metrics for open-ended text generation would require a fundamentally different evaluation framework. We agree this is a compelling direction for future work; it is already explicitly noted in our NLL limitations paragraph: "NLL does not take into account any improvements on semantically equivalent text."
>
> ---
>
> ### Scope of core claim
>
> > "The applicability boundary of the core claim remains under-specified relative to the evidence."
>
> We have tightened the core claim along several dimensions:
> - **Models**: All state-of-the-art language has been replaced with explicit references to "the large language models we study" throughout the Introduction and Discussion
> - **Syntax**: the syntax effect is now reported as context-dependent (4–20%) rather than a single "up to 20%" figure. What is learned beyond syntax remains unclear.
> - **Peer definition**: Conclusions are now scoped to our specific definition of peers (top-15 most-frequently @-mentioned accounts); the new "Peer definition" paragraph in Limitations explicitly acknowledges that conclusions might differ under alternative definitions.
> - **Sampling**: A new "Subject sampling" paragraph in Limitations acknowledges that subjects are drawn from a 30-day activity window, biasing toward more active users.

---

### Review · Reviewer_DrFH · 2026-04-29

**Summary Of Contributions:**

This paper asks how well LLMs can predict individual tweets on X. The authors test four models (1.5B to 70B) on 6.25M posts from 5,000+ users and their peers, comparing four context types (none, random, peer, user history). They find that prediction remains difficult even for the largest models; user history helps far more than peer posts, with ~20% of that gain coming from learned @-mentions and hashtags. Results replicate across prompting and finetuning setups and hold across demographic groups.

Key strengths:

1. This paper studies 5,000+ real users with actual social ties, far beyond prior theoretical or small-sample work.
2. Prompting and finetuning tell the same story, which strengthens confidence that the effect is real.

Key weaknesses:
1. The top-15 @-mentions ignores follows, retweets, likes, and DMs that also shape influence on the platform.
2. Locked into English tweets from one platform in one month (early 2023), so generalizability is unclear.

**Audience:**

Yes

**Audience Explanation:**

This work sits at the intersection of language modeling, computational social science, and AI ethics，three areas where TMLR readers are actively publishing. As LLMs get deployed in social media contexts (content generation, user profiling, misinformation detection), understanding their actual predictive limits matters for both researchers and practitioners. The paper's empirical scale and cross-model comparison make it a useful reference for anyone working on LLM evaluation or privacy-adjacent ML applications.

**Claims And Evidence:**

Yes

**Claims Explanation:**

This paper provides a valuable empirical test of how well LLMs can predict individual tweets, using 5,000+ users and four model families. The scale and cross-model comparison are real strengths, and the dual prompting/finetuning validation adds credibility.

However, defining peers solely through top-15 @-mentions is too narrow—Twitter influence also flows through follows, retweets, and likes, so the conclusion that "peers don't help much" is overstated. The one-month English-only data slice from early 2023 limits generalizability, and excluding images, links, and quote tweets misses a major part of actual platform usage. I also found the privacy-risk conclusion too strong: group-level unpredictability doesn't protect individually predictable users, a variance the appendix notes but the main text downplays.

**Requested Changes:**

1. Narrow the peer-influence claims. The paper currently frames its conclusion as "peers do not provide much predictive signal," but the experiments only test peers defined as top-15 @-mentions. Please revise to clarify that the finding applies specifically to this narrow definition, or broaden the peer selection to include follows, retweets, and likes.

2. Soften the privacy-risk conclusion. The claim that "concerns... are not supported by our findings" is too strong given that (a) the appendix shows substantial individual-level variance in predictability, and (b) the study does not assess non-linguistic profiling risks (location, political leaning, etc.). Group-level unpredictability does not imply individual safety.

---

> ### Author Response · Authors · 2026-05-07
> **Author Response**
>
> Thank you for your review! We are glad you found the scale of the study and the dual prompting/finetuning validation to be real strengths — we agree that the cross-method consistency is one of the paper's most important contributions. We hope the revisions below adequately address your concerns.
>
> ### Peer definition
>
> > "Narrow the peer-influence claims. The paper currently frames its conclusion as 'peers do not provide much predictive signal,' but the experiments only test peers defined as top-15 @-mentions. Please revise to clarify that the finding applies specifically to this narrow definition, or broaden the peer selection to include follows, retweets, and likes."
>
> We adopt your first suggestion. While follows, retweets, and likes may be informative, broadening peer selection along these dimensions is not straightforward: users typically follow hundreds to thousands of accounts (median: 302), making follows too diffuse; likes and retweets are similar in that respect and require less engagement than @-mentions, making them a noisier proxy of meaningful interaction. The new "Peer definition" paragraph in Limitations explicitly acknowledges that conclusions may differ under alternative operationalizations of the definition.
>
> ---
>
> ### Privacy risks
>
> > "Soften the privacy-risk conclusion. The claim that "concerns... are not supported by our findings" is too strong given that (a) the appendix shows substantial individual-level variance in predictability"
>
> We have made two changes to address this: the Introduction now reads "Concerns that large language models have made online speech broadly predictable are not supported by our findings, though we do observe substantial variability at the individual level," and the new Broader Impact section explicitly directs readers to the individual variability appendix and acknowledges that individual-level risks may differ considerably from population-level averages.
>
> > "(b) the study does not assess non-linguistic profiling risks (location, political leaning, etc.). Group-level unpredictability does not imply individual safety"
>
> We agree. The Broader Impact section now explicitly states that profiling risks may arise from non-linguistic signals — such as behavioral patterns, social graph structure, metadata, or images — which fall outside the scope of our text-based analysis.

---

### Author Response · Authors · 2026-05-07
**Revision Changes Summary**

We are grateful to the reviewers for their feedback, which has helped us improve our paper! The revised version makes our contributions more clear: we added more context about our learned syntax claims, we made limitations arising from sampling and peer definition explicit, and we added a Broader Impact section which clarifies the scope of our claims. Below is a summary followed by the concrete changes:

---

## Summary of Changes
1. **Syntax claim framing**: Added context dependency, highlighted 4-20% variation; Clarified scope; Difference in data composition
2. **Model framing**: Reframed intrinsic predictability claim; "State-of-the-art" removed
3. **Peer definition**: Added to Limitations; Removed text implying peer = close social ties
4. **Scope of conclusions**: Emphasize individual variability; added "Broader Impact" section; Scoped text-only findings
5. **Subject sampling**: Added to Limitations
6. **Grammar and citation corrections**
7. **Reference updates**

## Breakdown of Changes

### 1. Syntax claim framing
**Added context dependency, highlighted 4-20% variation**

Abstract
> "*Between 4\% and 20\%* of what is learned in-context is the use of @-mentions and hashtags, *depending on the type of context.*"

Introduction (contribution bullet 1)
> "*Between 4\% and 20\%* of the effect size can be attributed to the model learning to predict hashtags and @-mentions (i.e. syntax)*, depending on the type of context*"

**Clarified scope**

Results (Section 4.2 header + body)
> "*Base Models Learn Syntax, Among Other Things*"

> "Predicting @-mentions and hashtags correctly *is also a contributing factor* in the case of user and peer context."

> "The remaining 80–96% of the contextual gain is not explained by syntax; what exactly is learned beyond syntax remains an open question."

**Difference in data composition**

Results (Section 4.2)
> "*This variation partly reflects differences in syntax composition across context types*: random context contains ~35% more hashtags per tweet than peer context (0.27 vs.\ 0.20), while user context contains ~29% more @-mentions per tweet than peer context (0.99 vs.\ 0.77)."

### 2. Model framing

**Reframed intrinsic predictability claim**

Introduction (contribution bullet 1 header)
> "*LLMs struggle to predict online speech*"

Discussion (main finding)
> "online speech remains hard to predict *with the large language models we study*"

**"State-of-the-art" removed**

Introduction (closing paragraph)
> "limited even with *the large language models we study*"

Discussion (opening sentence)
> "using *four* large language models"

Discussion (summary sentence)
> "*the models we study* predict speech rather poorly"

### 3. Peer definition

**Added to Limitations**
Acknowledges alternative definitions of peers using follows/retweets/likes and provides justification for choosing @-mentions. Discusses limitations of peer selection based on @-mentions, and concedes that results may change under different operationalizations of the definition.

**Removed text implying peer = close social ties**

Setup (context list)
> "peer context: tweets from *accounts the user most frequently @-mentions*"

Setup (new note)
> "Note that peers defined in this way may include a *mix of genuine social contacts, public figures and institutional accounts.* We found that 16.7\% of peers belong to a verified account."

Introduction (contribution bullet 2)
> "posts from their *most frequently @-mentioned accounts*"

Abstract
> "models using posts from their *peers*"

### 4. Scope of conclusions

**Emphasize individual variability**

Introduction (closing paragraph)
> "Concerns that large language models have made online speech broadly predictable are not supported by our findings, *though we do observe substantial variability at the individual level.*"

**Added 'Broader Impact' section**
Final paragraph of Discussion removed; content moved to the new Broader Impact section which acknowledges:
- results inform but do not resolve questions about shadow profiling and impersonation;
- individual-level risks may differ from population-level averages (with pointer to individual variability appendix);
- demographic proxies are imperfect;
- profiling risks from non-linguistic signals not studied;
- need for updated conclusions as future models evolve.

**Scoped text-only findings**
Added sentence in Limitations section, 'External validity' paragraph noting that posts are often reactions to linked content, images, or ongoing interactions not captured by our text-based measure.

### 5. Subject sampling

**Added to Limitations**
Acknowledges that sampling from a 30-day activity window biases the sample toward more active users.

### 6. Grammar and citation corrections

- Full grammar pass; corrected `\citet`/`\citep` usage throughout

### 7. Reference updates
Now citing the published version of the papers:
- Xiao et al. (2022): EMNLP 2022
- Blodgett et al. (2016): EMNLP 2016
- Saunshi et al. (2021): ICLR 2021